# Type I Diabetes Pathoetiology and Pathophysiology: Roles of the Gut Microbiome, Pancreatic Cellular Interactions, and the ‘Bystander’ Activation of Memory CD8^+^ T Cells

**DOI:** 10.3390/ijms24043300

**Published:** 2023-02-07

**Authors:** George Anderson

**Affiliations:** CRC Scotland & London, Eccleston Square, London SW1V 1PG, UK; anderson.george@rocketmail.com

**Keywords:** type 1 diabetes, mitochondria, melatonin, *N*-acetylserotonin, aryl hydrocarbon receptor, TrkB, pancreatic beta cells, gut microbiome, circadian, treatment

## Abstract

Type 1 diabetes mellitus (T1DM) arises from the failure of pancreatic β-cells to produce adequate insulin, usually as a consequence of extensive pancreatic β-cell destruction. T1DM is classed as an immune-mediated condition. However, the processes that drive pancreatic β-cell apoptosis remain to be determined, resulting in a failure to prevent ongoing cellular destruction. Alteration in mitochondrial function is clearly the major pathophysiological process underpinning pancreatic β-cell loss in T1DM. As with many medical conditions, there is a growing interest in T1DM as to the role of the gut microbiome, including the interactions of gut bacteria with *Candida albicans* fungal infection. Gut dysbiosis and gut permeability are intimately associated with raised levels of circulating lipopolysaccharide and suppressed butyrate levels, which can act to dysregulate immune responses and systemic mitochondrial function. This manuscript reviews broad bodies of data on T1DM pathophysiology, highlighting the importance of alterations in the mitochondrial melatonergic pathway of pancreatic β-cells in driving mitochondrial dysfunction. The suppression of mitochondrial melatonin makes pancreatic β-cells susceptible to oxidative stress and dysfunctional mitophagy, partly mediated by the loss of melatonin’s induction of PTEN-induced kinase 1 (PINK1), thereby suppressing mitophagy and increasing autoimmune associated major histocompatibility complex (MHC)-1. The immediate precursor to melatonin, *N*-acetylserotonin (NAS), is a brain-derived neurotrophic factor (BDNF) mimic, via the activation of the BDNF receptor, TrkB. As both the full-length and truncated TrkB play powerful roles in pancreatic β-cell function and survival, NAS is another important aspect of the melatonergic pathway relevant to pancreatic β-cell destruction in T1DM. The incorporation of the mitochondrial melatonergic pathway in T1DM pathophysiology integrates wide bodies of previously disparate data on pancreatic intercellular processes. The suppression of *Akkermansia muciniphila*, *Lactobacillus johnsonii*, butyrate, and the shikimate pathway—including by bacteriophages—contributes to not only pancreatic β-cell apoptosis, but also to the bystander activation of CD8^+^ T cells, which increases their effector function and prevents their deselection in the thymus. The gut microbiome is therefore a significant determinant of the mitochondrial dysfunction driving pancreatic β-cell loss as well as ‘autoimmune’ effects derived from cytotoxic CD8^+^ T cells. This has significant future research and treatment implications.

## 1. Introduction

Type 1 diabetes mellitus (T1DM) is a chronic condition arising from a failure of pancreatic β-cells to produce adequate insulin, often as a consequence of pancreatic β-cell destruction, thereby dysregulating circulating glucose level homeostasis. T1DM is classified as an immune-mediated disorder, with raised levels of autoantibodies. The risk of progressing to T1DM is negatively correlated with age [1]. A number of risk factors are evident, including genetic and viral, with pathoetiology predominantly driven by environmental and epigenetic factors. T1DM most commonly emerges in childhood and adolescence, although it can also arise in adulthood. Whilst screening can lead to an earlier diagnosis, the majority of T1DM diagnoses are made when most pancreatic β-cells and their function are significantly disturbed. There is no known treatment that tackles the primary site of T1DM pathoetiology, namely pancreatic β-cell loss and dysfunction. Rather, treatment is primarily targeted to monitoring and regulating blood glucose levels. As T1DM is associated with functional disturbances in many organs, it is linked to an elevated risk of numerous other medical conditions, including dementia [2,3], cardiovascular disorders, kidney disorders, retinopathy, neuropathy, hypertension, lung disorders, obesity, and amyotrophic lateral sclerosis (ALS) [4], as well as bacterial and fungal infections [5,6]. As all these comorbidities show alterations in the gut microbiome and gut permeability [7,8], the gut dysbiosis evident in T1DM is an important mediator of its pathophysiological associations of these comorbidities [9].

This article reviews wide bodies of data on the biological underpinnings of T1DM, highlighting the powerful role of the mitochondrial melatonergic pathway in pancreatic β-cells in integrating previously disparate bodies of data on T1DM, with crucial effects mediated via alterations in the gut microbiome. Future research and treatment implications are indicated.

## 2. Classical Type 1 Diabetes Mellitus Pathoetiology

T1DM arises from the dysregulation of pancreatic β-cell insulin production and insulin effects. Pancreatic β-cell function and insulin release are intimately linked to variations in blood glucose. At low/basal levels of blood glucose, the ATP-sensitive potassium (K_ATP_) channels in pancreatic β-cells remain open, leading to maintained membrane hyperpolarization coupled to Ca^2+^ channel closure, thereby inhibiting insulin secretion by pancreatic β-cells [10]. When blood glucose concentration levels rise, glucose is taken up by the glucose transporter (GLUT)1 into pancreatic β-cells where it is quickly phosphorylated by glucokinase and then converted to pyruvate. The conversion of pyruvate to acetyl-CoA by the pyruvate dehydrogenase complex (PDC) increases ATP production by the tricarboxylic acid (TCA) cycle and mitochondrial oxidative phosphorylation (OXPHOS), resulting in K_ATP_ channel closure, plasma membrane depolarization, and the opening of voltage-dependent Ca^2+^ channels in association with insulin vesicle exocytosis. Mitochondrial function is therefore a crucial aspect of pancreatic β-cell function and plasticity of response to variations in circulating glucose [10]. 

Islet amyloid polypeptide (amylin) is simultaneously secreted by pancreatic β-cells along with insulin. Amylin accumulates in diabetes, forming amyloid deposits in the pancreas that are a relevant aspect of T1DM and T2DM pathophysiology, driving pancreatic β-cell dysfunction and apoptosis, as well as contributing to islet transplantation failure [11,12]. Amylin fibrillation follows amylin over-production in association with insulin resistance and hyperinsulinemia, which triggers a nucleation-dependent self-assembly of amylin into intracellular or extracellular amyloid deposits [13]. Amylin fibrillation is proposed to parallel other amyloidosis, such as amyloid-β in dementia, in driving pancreatic β-cell apoptosis and may therefore be an important treatment target in T1DM [13], including in relation to the ‘autoimmune’ aspects of T1DM. Amylin, Ca^2+^ dysregulation, reactive oxygen species (ROS), endoplasmic reticulum alterations, and metabolic dysregulation have all been proposed to drive post-translational protein modifications that underpin the autoimmune-like response in T1DM [11].

T1DM pathophysiology is classically described as a T cell-mediated autoimmune disorder, whereby predominantly CD8^+^ T cells become autoreactive, leading to the destruction of pancreatic β-cells. A failure in the capacity of regulatory T cells (Treg) to suppress such processes is also implicated. Elevated levels of major histocompatibility complex (MHC) class 1 molecules on pancreatic β-cells drive the autoreactivity responses of CD8^+^ T cells. Such autoreactive CD8^+^ T cells usually undergo negative selection in the thymus, which seems to be altered/impaired in T1DM due, at least partly, to an allelic variant of the nuclear factor kappa-light-chain-enhancer of activated B cells (NF-κB) modulator, NF-κB-inhibitor Delta (Nfkbid), which is an atypical NF-κB inhibitor and an important regulator of B cell immunity across a host of medical conditions [14]. Modulation of Nfkbid thymic expression levels indicates that it may act to regulate levels of both autoreactive CD8^+^ T cells and Treg [15,16].

T1DM susceptibility is linked to several genetic and environmental factors. Familial transmission is apparent in about 10% of T1DM patients, with the human leukocyte antigen complex (HLA) located on chromosome 6 showing the strongest familial association [17]. Numerous single nucleotide polymorphisms (SNPs) have been linked to T1DM risk, including P53 [18], the NF-κB modifying gene, small ubiquitin-related modifier 4 (SUMO4) [19], and many other genes [20] as well as HLA haplotypes [21]. Many T1DM susceptibility genes act to regulate mitochondrial function and mitophagy in pancreatic β-cells, including Clec16a [22], highlighting the importance of optimal mitochondrial function and OXPHOS [23] in the regulation of pancreatic β-cell function and insulin regulation. 

T1DM environmental risk factors include the consumption of a Western-type diet, with a number of ligands for the receptor for advanced glycation end-products (RAGE) contained within many such foods [24]. These authors propose that RAGE activation leads to the activation and proliferation of islet infiltrating CD8^+^ and CD4^+^ T cells, coupled to Treg suppression, thereby impacting on the patterned immune response that contributes to pancreatic β-cell injury, with effects mediated via NF-κB upregulation, coupled to increased ROS and oxidative stress [24]. RAGE has numerous other ligands, including high-mobility group box (HMGB)1 and S100/calgranulin family members, such as S100A9, as well as amyloid-β and pre-fibrillar amylin aggregates [25,26]. Preclinical data show that the administration of soluble RAGE dramatically suppresses T1DM via Treg upregulation, and associated inhibition of conventional T cell division [27]. Such data would implicate a significant clinical impact of RAGE ligands that can be suppressed by soluble RAGE, as well as melatonin [28]. The antioxidant, anti-inflammatory, circadian, and mitochondrial optimizing effects of melatonin are important aspects of T1DM, whilst the immediate precursor of melatonin, *N*-acetylserotonin (NAS) may also be a crucial regulator of pancreatic β-cell survival, via its capacity to activate the brain-derived neurotrophic factor (BDNF) receptor TrkB, as shown in Figure 1. The role of the melatonergic pathway in T1DM is detailed in Section 5. 

However, many of the genetic and epigenetic susceptibility factors for T1DM may ultimately act on mitochondrial function in pancreatic β-cells [23], immune cells, and other cells across the body. 

## 3. Wider T1DM Pathophysiology

A wide array of diverse signaling pathways, processes, and factors are associated with T1DM pathoetiology and pathophysiology.

The inability to prevent pancreatic β-cell loss and to induce regeneration, coupled with the complexity of immune-mediated processes underpinning autoimmune-like changes, has generated the investigation of a wide range of possible pathophysiological processes and factors in T1DM, including the aryl hydrocarbon receptor (AhR) [29], toll-like receptor (TLR)4 activation [30], NF-κB [31], yin yang (YY)1 [32], the melatonergic pathway [33], P53 [18], circadian dysregulation [34], and gut dysbiosis/permeability [35]. 

The AhR is predominantly expressed in a complex with other proteins in the cytoplasm, with its activation by endogenous and exogenous ligands leading to its translocation to the nucleus, where it forms a dimer that upregulates genes containing the xenobiotic response element. The AhR has complex effects that are partly dependent upon the specific AhR-activating ligand, cell type, and concurrent wider cellular processes [36]. The AhR is also expressed on the mitochondrial membrane where it can regulate Ca^2+^ influx via the voltage-dependent anion channel (VDAC)1 [37]. The AhR has diverse effects in different cells and tissues relevant to T1DM pathophysiology, with AhR activation leading to suppressed function and cytotoxic capacity in CD8^+^ T cells and natural killer (NK) cells [38], the maintenance of the gut barrier [39], and pancreatic β-cell regulation [40]. The diverse, and sometimes contrasting, effects of AhR activation in different cells and tissues by various ligands complicate its incorporation into T1DM pathophysiology.

Western diet-driven increases in palmitate and lipopolysaccharide (LPS) can synergistically damage pancreatic β-cells via TLR4 activation, including via enhanced pancreatic β-cell ceramide and suppressed sphingosine-1-phosphate (S1P) levels [30]. The role of TLR4 signaling in pancreatic β-cell dysfunction has been widely replicated [41], with factors acting to regulate TLR signaling, including miR-383 and TLR4 knockout, ameliorating high fat/sugar diet-induced β-cell dysfunction, as mostly assessed in T2DM [42,43]. The protection afforded by melatonin in pancreatic β-cells is at least partly mediated via TLR2/4 level suppression [44]. Such data also indicate the relevance of increased gut permeability to pancreatic β-cell dysfunction via raised circulating LPS levels [45], with the association of palmitate with pancreatic β-cell dysfunction partly mediated via increased gut permeability [46]. 

LPS activation of TLR4 induces the transcription factors, NF-κB and YY1, in many cell types, including leukocytes, macrophages, microglia, and astrocytes [36,47]. The induction of pro-inflammatory processes in pancreatic β-cells is intimately linked to NF-κB upregulation [31,48,49]. The downstream consequences of NF-κB upregulation are associated with a diverse array of factors, including non-steroidal anti-inflammatory drug activated gene-1, growth differentiation factor-15 (NAG-1/GDF15) [48], or intercellular adhesion molecule (ICAM)-1 [31]. YY1 knockout in pancreatic β-cells leads to rapid hyperglycemia onset, impaired glucose tolerance, and suppressed pancreatic β-cell mass, both in neonates and adult murine models [32]. These authors showed YY1 to bind the enhancer regions in exon 2 of Ins1 and Ins2, activating pro-insulin and insulin transcription and thereby insulin production from pancreatic β-cells [32]. 

Such data on NF-κB and YY1 indicate contrasting effects in T1DM pathophysiology. However, it is important to note that both transcription factors can induce the melatonergic pathway, as shown in different cell types [50,51]. Variation in the capacity of these transcription factors to upregulate the melatonergic pathway is of some importance to T1DM pathophysiology, given that melatonin protects and optimizes pancreatic β-cell function and insulin regulation at nM levels [33,52], including via the optimization of mitochondrial function. This may be of relevance to data showing that the induction and activation of NF-κB may afford protection against pancreatic β-cell loss in T1DM models, which the authors attribute to NF-κB induction of miR-150, thereby preventing T1DM-associated inflammation and pancreatic β-cell apoptosis [53]. As NF-κB is associated with the induction of the melatonergic pathway in several cells so far investigated [51,54], it is of note that the most commonly used preclinical T1DM model, the streptozotocin-induced T1DM model, suppresses local melatonin production and melatonergic pathway activity, as shown in the retina [55]. This could indicate that the suppression of the mitochondrial melatonergic pathway may be a significant factor in pancreatic β-cell apoptosis in clinical T1DM, which is elaborated upon in Section 7. 

Other intracellular signaling processes and transcription factors are associated with T1DM, including SNPs in, and upregulation of, the classical tumor suppressor and transcription factor, P53 [18]. However, although P53 is upregulated during apoptosis in pancreatic β-cell in the course of T1DM and T2DM, recent data indicate that P53 is not essential to pancreatic β-cell loss [56]. Using pancreatic β-cell specific P53 knockout, these authors showed pancreatic β-cell specific P53 knockout failed to ameliorate insulin secretion and glucose tolerance, as well as failing to raise pancreatic β-cell numbers in an array of genetic, dietary, and pharmacological models of T1DM and T2DM [56]. These authors suggest that the induction of poly [ADP-ribose] polymerase 1 (PARP-1) may be a more relevant mediator of pancreatic β-cell loss in T1DM models [56], by decreasing NAD^+^ availability, thereby suppressing the sirtuin-induced PDC and mitochondrial OXPHOS. 

Circadian dysregulation, both genetic and environmental, is closely associated with metabolic syndrome and T2DM, involving the uncoupling of OXPHOS, ATP production, and glucose-stimulated insulin secretion [34], mediated—in part—via the suppression of the circadian gene, Bmal1, and the endogenous antioxidant inducing transcription factor, Nrf2 [34]. These changes can be reversed by melatonin [57,58], highlighting the importance of pineal melatonin in the circadian optimization of pancreatic β-cell function. Circadian dysregulation is also intimately linked to a diverse array of T1DM symptomatology [59], including nocturnal non-dipping blood pressure increasing kidney disease [60], cardiac autonomic neuropathy [61], platelet morphology [62], microvascular complications [63], and patterned immune activity [64], whilst circadian variation in basal insulin requirement can be an early marker of autoimmune polyendocrine syndromes in T1DM [65]. Such data highlight the role of alterations in the circadian rhythm in T1DM pathophysiology. 

The growing appreciation of the role of the gut microbiome and gut permeability across a host of diverse medical conditions is also highly relevant in T1DM [66,67]. The role of microbiota in T1DM and T2DM is highlighted by data showing the impact of the maternal endometrial and/or vaginal microbiomes of diabetic mothers, including in gestational diabetes mellitus, which can act through epigenetic mechanisms to increase T2DM—and possibly T1DM—risk in the offspring [67]. Such data highlight the importance of microbiota in the regulation of metabolism and the likelihood of prenatal priming of later T1DM related processes in pancreatic β-cells. This requires future investigation. 

Recent data causal models the beneficial effects of human umbilical cord mesenchymal stem cell vesicles containing exenatide as being mediated not only directly in pancreatic β-cells but also via alterations in the gut microbiome/permeability [68]. The beneficial effects of *Lactobacillus johnsonii* strain N.6 nanovesicles in T1DM are proposed to be mediated via the upregulation of AhR ligands and AhR activation, with effects in both pancreatic β-cells as well as in macrophages, which are induced into a M2b-like phenotype [69]. A number of preclinical studies over the past decade have shown the beneficial effects of *Lactobacillus johnsonii* in delaying T1DM onset, which is proposed to be mediated by a number of processes, including suppression of Th17 cells, increasing intestinal crypt Paneth cell numbers, and suppressing gastro-intestinal caspase-1 induction [70,71,72], whilst also decreasing the kynurenine/tryptophan ratio, increasing cytotoxic CD8^+^ T cells, and changing the patterned immune response, as shown in healthy human volunteers [73]. Whether *Lactobacillus johnsonii*, and other bacteria, in the gut drive the induction of AhR ligands (such as indole-3-propionate) will be important to determine. *Lactobacillus johnsonii* also increases the short-chain fatty acids, butyrate, propionate, and acetate, indicating wider effects of gut microbiome-derived products, including butyrate’s epigenetic effects as a histone deacetylase (HDAC) inhibitor [74]. HDACi, via PKA upregulation and tryptophan hydroxylase (TPH)1 induction, derepresses serotonin production, thereby potentiating pancreatic β-cell function [75]. As butyrate optimizes mitochondrial function with effects that involve the upregulation of the melatonergic pathway, as shown in intestinal epithelial cells [76], alterations in the capacity to upregulate the melatonergic pathway in pancreatic β-cells will be important to determine. It clearly requires investigation as to the relevance of gut microbiome-derived butyrate, including as regulated by *Lactobacillus johnsonii*, in the pathoetiology of T1DM and the importance of a functional tryptophan–melatonin pathway in pancreatic β-cells. The full effects of butyrate require the capacity of a cell to upregulate the mitochondrial melatonergic pathway [76]. 

Importantly, the gut microbiome is comprised of not only bacteria, but also fungi and viruses, including enteroviruses and bacteriophages, with all these groupings showing changes at the initiation of pediatric T1DM [77,78]. Investigations on the gut microbiome in T1DM, versus controls, have focused on changes in gut bacteria, showing elevations in *Prevotella copri* and *Eubacterium siraeum*, with relative attenuation of *Firmicutes bacterium* and *Faecalibacterium prausnitzii* [79] in T1DM patients. Other studies have investigated a wider range of changes in the gut microbiome, indicating no significant differences in α-diversity between T1DM and controls [80]. However, these authors found T1DM patients to have 43 bacterial taxa significantly depleted and 37 bacterial taxa significantly enriched [80]. This study also found disease duration and glycated hemoglobin (HbA1c) to explain a significant part of the gut microbiome variation in T1DM, whilst neuropathy and macrovascular complications were significantly linked to variations in several microbial species [80]. However, as noted by the authors of these studies [79,80], and other studies [81], the mechanistic links to pancreatic β-cell loss and wider T1DM pathophysiology remain to be determined. 

Limited data on bacteriophages in the pathoetiology of T1DM indicate that amyloid-producing *Escherichia coli* (*E. coli*), *E. coli* phages, and bacteria-derived amyloid may be involved in the early stages of T1DM pathoetiology, as indicated in data derived from children at high risk of T1DM [82]. This study and other data indicate that changes in the gut virome may precede the initial signs of T1DM, and therefore be of relevance to T1DM pathoetiology [83]. The causative relationship has still to be determined. However, such data may indicate that the gut virome may be more relevant to T1DM pathoetiology than gut bacteria, which tend to show diversity only after the emergence of T1DM. Enteroviruses are one of the major environmental triggers of childhood-onset T1DM, with recent data indicating that enteroviruses may also be an important trigger in adult-onset T1DM [84]. Overall, data indicate an interrelatedness of the gut microbiota, metaproteome, and virome that is relevant to T1DM onset, as investigated in young children, with a functional remodeling of the gut microbiota accompanying islet autoimmunity [77]. An initial bacteriophage or enterovirus impact on gut microbiome diversity seems followed by a decrease in butyrate-producing bacteria, with consequences for mitochondrial function systemically. As to whether the suppressed butyrate and/or other gut bacteria products are drivers of changes in pancreatic β-cells, either directly or indirectly suppressing the mitochondrial melatonergic pathway in pancreatic β-cells requires investigation. 

Interestingly, *Candida albicans* fungi is elevated in T1DM, including at the time of initial presentation [85]. As *Lactobacillus johnsonii* can eliminate *Candida albicans* fungi from the gut [86], the clinical benefits of *Lactobacillus johnsonii* are likely to include alterations in the wider gut microbiome. However, it still requires clarification as to whether the gut is a primary site of change or whether the emergence of fungal infections is driven by a suppressed anti-fungal immune response [87], which may also involve alterations in the gut microbiome [67]. Two fatty acids produced by *Lactobacillus johnsonii* (and *Bacteroides thetaiotaomicron*), namely oleic acid and palmeic acid, mediate many of *Lactobacillus johnsonii* benefits, including within the gut and immune cells, indicating that gut microbiome-associated regulation of the immune response is a relevant aspect of T1DM pathoetiology [88]. Future research will have to clarify the relevance of *Lactobacillus johnsonii* specific effects in T1DM, including in interaction with *Candida albicans* fungal infection.

## 4. Pancreatic β-Cell Mitochondria and Metabolism

It has long been appreciated that mitochondrial function is a crucial determinant of pancreatic β-cell function, with heightened glucose concentrations stimulating mitochondrial OXPHOS, thereby stimulating intracellular ATP and ADP levels, ultimately leading to insulin secretion. Glucose-derived pyruvate, rather than exogenous lactate or pyruvate, is necessary for pancreatic β-cell to produce and release insulin [89]. These authors showed that various miRNAs are expressed in pancreatic β-cells to limit the uptake of lactate and pyruvate, thereby restricting other metabolic drivers [89]. The mitochondrial metabolism of pyruvate is crucial for determining insulin secretion, with pyruvate taken up into mitochondria by the mitochondrial pyruvate carriers (Mpc1 and Mpc2) [90]. Mitochondrial pyruvate and associated OXPHOS drive insulin secretion by both K_ATP_ channel-dependent as well as -independent pathways [90]. 

Mitochondrial dysfunction, driven by an array of specific factors investigated in single studies, alters the associations of glucose and insulin secretion. The loss of mitochondrial transcription factor B1 (TFB1M) leads to mitochondrial dysfunction and T2DM pathogenesis [91], as does the ribosomal RNA (rRNA) methyltransferase homolog of TFB1M, namely dimethyladenosine transferase 1 homolog (DIMT1) [92]. These authors showed that DIMT1 knockout suppresses OXPHOS-derived ATP and protein synthesis in association with suppressed insulin secretion [92]. As with many relevant cells across diverse medical conditions, alterations in mitochondrial function are a crucial aspect of pancreatic β-cell changes in T1DM.

Recent work indicates the importance of wider pancreatic β-cell metabolism in driving metabolic- and glucose-induced insulin secretion, including via the nutrient sensing mitochondrial guanosine-5′-triphosphate (mtGTP) cycle, involving the synthesis of phosphoenolpyruvate (PEP), a high-energy metabolite, which integrates the tricarboxylic acid (TCA) cycle and anaplerosis with glucose-stimulated insulin secretion [93]. PEP and pyruvate kinase (PK) play important roles in how nutrient-stimulated K_ATP_ channel closure/opening regulates insulin secretion [94]. ATP closes K_ATP_ channels to stimulate insulin secretion, including from ATP derived PEP stimulated plasma membrane PK, with contrasting effects of PKm1 vs. PKm2 on the glycolytic and mitochondrial sources of PEP, in the regulation of K_ATP_ channel opening and closure [91]. Notably, the regulation of PEP is intimately linked to the TCA cycle [93,94].

Recent data indicate that mitochondrial ‘subtypes’ may be evident in healthy controls and diabetes patients [95]. These authors showed mitochondrial ‘subtypes’ could be classed according to mtDNA- and nuclear DNA-encoded OXPHOS genes [95]. In a study of global miRNA expression in T1DM patients vs. controls, 41 miRNAs were differentially expressed, of which 34% targeted mitochondrial genes, with over 80% (33/41) targeting nuclear genes involved in mitochondrial metabolism [96]. Although not specific to pancreatic β-cells, such data highlight the importance of alterations in mitochondrial metabolism in T1DM pathophysiology. As to whether the mitochondrial ‘subtypes’ described survive further investigation remains to be determined [95], especially when alterations in the regulation of the mitochondrial melatonergic pathway are included. However, the data in this study do emphasize how an array of factors and processes may associate with changes in mitochondrial function [95].

A plethora of studies have highlighted the role of pancreatic β-cell mitochondria as a crucial aspect of T1DM pathophysiology. Recent work indicates that the mitochondrial melatonergic pathway may be a core aspect of mitochondrial function, with relevance across a host of diverse medical conditions, including Alzheimer’s disease [97,98], multiple sclerosis [99,100], glioblastoma [101,102], breast cancer [103,104], depression [105,106], and amyotrophic lateral sclerosis [7,107,108]. Integrating the melatonergic pathway into the biological underpinnings of T1DM allows for the inclusion of previous disparate bodies of data, providing a conceptualization of T1DM that has treatment and future research implications.

## 5. Melatonergic Pathway and T1DM

Given its antioxidant, anti-inflammatory, antinociceptive, and mitochondria-optimizing effects, melatonin has ubiquitous benefits across most medical conditions, including when applied to T1DM patients and preclinical models. In a preclinical study of the effects of melatonin alone or combined with the dipeptidyl peptidase IV, sitagliptin, combination therapy induced mouse pancreatic β-cell regeneration under glucotoxic stress, which was replicated in the human islet transplant mouse model. Combination therapy also induced pancreatic β-cell proliferation, reduced fasting blood glucose levels, and enhanced plasma insulin levels and glucose tolerance [109]. When used alone, melatonin reduced pancreatic β-cell apoptosis [109]. Melatonin also prevents kidney disorders, bone loss, retinal dysfunction, and cognitive dysfunction evident in streptozotocin-induced T1DM in preclinical models [110]. Suppressed pineal melatonin is often evident in T1DM patients [111], indicating a suppressed capacity of circadian melatonin to dampen inflammation and reset immune cell mitochondrial metabolism. Most of the pathophysiological changes in T1DM, including oxidative stress, suboptimal mitochondrial function, increased apoptotic pathways, and immune dysregulation are all suppressed by melatonin [112]. Although the tryptophan–melatonin pathway is potentially of crucial relevance in T1DM, it has been dramatically under-investigated in the research and treatment of T1DM. 

The tryptophan–melatonin pathway is evident in cells across body organs and tissues, with relevance to a host of diverse medical conditions, including depression [36], amyotrophic lateral sclerosis [7], cancers [113] and dementia [97]; see Figure 1. Most tryptophan is derived from dietary sources, although the shikimate pathway in the human gut microbiome is a relevant provider of aromatic amino acids, including tryptophan, as well as tyrosine and phenylalanine [7]. The shikimate pathway is proposed below to be a significant aspect of T1DM pathophysiology. Tryptophan can be converted by tryptophan decarboxylase in the gut to tryptamine, which activates the gut AhR to maintain the gut barrier [114]. The uptake of tryptophan into the circulation allows it to be taken up by cells, usually via the large amino acid transporter (LAT)1, where it is converted to serotonin (5-HT) by tryptophan hydroxylase (TPH), either TPH2 (mostly in the CNS) or TPH1 in other body organs. TPH2, and possibly TPH1, requires stabilization by 14-3-3ε, which enable TPH to convert tryptophan to 5-HT. Cellular 5-HT can also be provided from serotonergic neuronal inputs as well as from circulating platelets. Once available in cells, 5-HT can be converted by aralkylamine *N*-acetyltransferase (AANAT) to *N*-acetylserotonin (NAS). As with TPH2, AANAT requires stabilization by 14-3-3, likely 14-3-3ζ, whilst AANAT also requires acetyl-CoA as a necessary co-substrate. Factors affecting the availability of these 14-3-3 isoforms and acetyl-CoA will therefore limit the cell’s capacity to induce the melatonergic pathway. The melatonergic pathway is mostly expressed within mitochondria, allowing the melatonergic pathway to be intimately associated with mitochondrial metabolism. This is exemplified by the requirement of acetyl-CoA for the conversion of 5-HT to NAS; see Figure 2, which shows how the tryptophan–melatonin pathway may be integrated into wider T1DM pathophysiology.

The process of OXPHOS requires the disinhibition of the pyruvate dehydrogenase complex (PDC) by mitochondria-located sirtuin-3 or the circadian gene, brain, and muscle ARNT-Like 1 (Bmal1), allowing PDC to convert pyruvate to acetyl-CoA. As well as providing acetyl-CoA for the conversion of 5-HT to NAS, acetyl-CoA is also necessary to optimize ATP production by the tricarboxylic acid (TCA) cycle and OXPHOS. The induction of the mitochondrial melatonergic pathway is therefore intimately linked to mitochondrial metabolism, as is insulin production. As well as providing direct antioxidant effects, melatonin will have a plethora of intracrine, autocrine, and paracrine effects, including providing a ‘film’ that coats the mitochondrial outer membrane, regulating membrane fluidity, and regulating the expression of mitochondrial membrane channels, receptors, and transporters [115] as well as inducing endogenous antioxidants and enzymes, such as catalase and glutathione (GSH) [116]. The capacity of a cell’s mitochondria to induce the melatonergic pathway is therefore a significant determinant of their capacity to resist challenge.

### Melatonergic Pathway and Wider T1DM Pathophysiology

As well as being blocked by the inhibition of tryptophan conversion to 5-HT or the uptake of serotonin from other sources, the melatonergic pathway is subject to modulation by a number of processes that can have significant consequences for cellular function and intercellular interactions. Activation of the AhR, metabotropic glutamate receptor (mGluR)5, or purinergic P2Y1r drives the ‘backward’ conversion of melatonin to NAS via *O*-demethylation. Although NAS has some overlapping effects with melatonin, e.g., both are antioxidants, NAS is uniquely a brain-derived neurotrophic factor (BDNF) mimic via its activation of the BDNF receptor, TrkB [117]. Given the importance of BDNF to the functioning of pancreatic β-cells, factors regulating the NAS/melatonin ratio, such as the AhR, P2Y1r, and mGluR5, may be highly relevant in the regulation of pancreatic β-cell responses, as well as in the pathophysiology of T1DM. The suppression of the melatonergic pathway will therefore have dramatic consequences for AhR, mGluR5, and P2Y1r activation. Regulation of the tryptophan–serotonin–melatonin pathway will be relevant not only in pancreatic β-cells, but also to the interactions of pancreatic β-cells with other cells in the pancreatic islets, as well as with circulating immune cells and platelets. 

The raised levels of pro-inflammatory cytokines in T1DM will induce indoleamine 2,3-dioxygenase (IDO) and tryptophan 2,3-dioxygenase (TDO), with the latter also induced by stress-associated hormones, including cortisol. IDO and TDO convert tryptophan to kynurenine, thereby not only suppressing tryptophan availability for the tryptophan–melatonin pathway, but also providing kynurenine and kynurenic acid as ligands for the AhR, thereby increasing the NAS/melatonin ratio, coupled to enhanced TrkB activation by NAS and suppression of melatonin availability. Some of the complex—and sometimes contrasting—effects of AhR activation seem to arise, at least partly, from its capacity to upregulate NAS and TrkB activation. As noted, TrkB activation is an important trophic and pro-survival signal in pancreatic β-cells, including from muscle activity [118]. The suppression of the melatonergic pathway will therefore have consequences for the impact of other receptors and intercellular signaling processes; see Figure 2.

Gut dysbiosis and gut permeability are linked to a wide range of diverse medical conditions. Two of the major factors arising from gut dysbiosis and gut permeability are decreased butyrate and increased circulating LPS, respectively. Both of these gut microbiome-derived factors are intimately intertwined with variations in the melatonergic pathway; see Figure 3. LPS activation of TLR4/NF-κB-YY1 drives inflammatory processes in many cells, including pancreatic β-cells [41]. The capacity of NF-κB and YY1 to upregulate the melatonergic pathway will determine the level and duration of inflammatory signaling via intracrine and autocrine melatonin effects, whilst the paracrine effects of released melatonin will dampen local inflammatory processes. As noted, the suppression of gut microbiome-derived butyrate will attenuate butyrate’s induction of sirtuin-3/PDC/OXPHOS and the melatonergic pathway, whilst also increasing the HDAC potentiation of YY1-induced transcriptions. The consequences of gut dysbiosis and gut permeability are therefore intimately linked to alterations in the regulation of the melatonergic pathway across central and systemic cells, including pancreatic β-cells. This has implications for the wider signaling proposed to underpin pancreatic β-cell loss, including as driven by HMGB1 [119]. See Figure 3. 

HMGB1 activation of the TLR2/TLR4/NF-κB pathway has been proposed to contribute to the loss of pancreatic β-cells, with sodium butyrate inhibiting pancreatic HMGB1 and NF-κB p65 protein expression in the streptozotocin-induced T1DM model [119]. However, other data indicate that HMGB1 may provide some protection to pancreatic β-cells [120]. As to whether this seeming disparity arises from variability in the capacity of NF-κB and YY1 to induce the melatonergic pathway will be important to determine, especially as streptozotocin has been shown to suppress the melatonergic pathway in other cell types [55]. The capacity of butyrate to protect pancreatic β-cells in the streptozotocin model [119] could indicate that its protection may be at least partly afforded by its capacity to upregulate the melatonergic pathway [76,121]. 

Recent data implicate BDNF activation of TrkB in driving trophic and protective effects in pancreatic β-cells [122]. This would indicate that the autocrine and paracrine effects of released NAS would similarly have protective effects in pancreatic β-cells. The role of BDNF and TrkB in pancreatic β-cells and T1DM may therefore be intimately linked to the mitochondrial melatonergic pathway. TrkB signaling is complicated by the existence of full-length (TrkB-FL) and truncated (TrkB-T1) forms. Generally, TrkB-FL is associated with trophic signaling, whilst TrkB-T1 is linked to cellular dysfunction and apoptosis arising from lost trophic signaling, as exemplified in motor neuron loss in amyotrophic lateral sclerosis [7]. However, this does not seem to be the case in murine pancreatic β-cells [118]. These authors showed that TrkB.T1 knockout leads to impaired glucose tolerance and insulin secretion in mice [118], with BDNF acting on TrkB.T1 in pancreatic β-cells to trigger calcium release from intracellular stores that enhances glucose-induced insulin secretion. Interestingly, muscle activity-induced BDNF also increases insulin release, linking muscle-derived BDNF to enhanced glucose metabolism in response to exercise [118]. 

Such data on the role of BDNF in pancreatic β-cells [118,122] may have a number of implications for alterations in the tryptophan–melatonin pathway in T1DM. The relevance of different cellular sources of NAS, and NAS-induced BDNF [123], will be important to determine in pancreatic islets, not only from pancreatic β-, α-, δ-, and ε-cells, but also from other cells in the pancreatic islet microenvironment, including macrophages and other immune cells, as well as circulating cells such as platelets. Overall, data on BDNF and TrkB in T1DM may be another ready link to the relevance of the tryptophan–melatonin pathway.

The role of NAS in T1DM pathophysiology is given some indirect support. IDO suppression in pancreatic β-cells is one of the last events to occur before cellular apoptosis [124]. This would indicate that the pro-inflammatory cytokine-induced IDO-kynurenine-AhR activation, increasing NAS for TrkB-T1 (or TrkB-FL), may be providing pro-survival benefits under challenge, with released kynurenine, via AhR activation on CD8^+^ T cells, expected to suppress their cytolytic capacity. The three known inducers of NAS from melatonin, namely the AhR, P2Y1r, and mGluR5, can potentiate insulin release from pancreatic β-cells [125,126]. The activation of the cystine–glutamate antiporter (system χ_c_) may be relevant to glutamate effects in pancreatic β- and α-cells, whereby the demand for GSH production will drive the release of glutamate, which can act on α-cells to activate the AMPA/kainate receptors, thereby enhancing glucagon responses in the prevention of hypoglycaemia [127]. In pancreatic β-cells, system χ_c_ overexpression prevents 2-deoxy-d-ribose-induced pancreatic β-cell damage, with relevance to both T1DM and T2DM [128]. The authors attribute the benefits of system χ_c_ overexpression to enhanced GSH availability. However, as to whether released glutamate has autocrine and paracrine effects that activate the AMPA/kainite receptors in α-cells as well as the mGluR5 in pancreatic β-cells (thereby increasing NAS at TrkB) will be important to determine. This is another example of how disparate data on T1DM may be better integrated by incorporation of the melatonergic pathway. 

Fibroblast growth factor (FGF)-2 also upregulates system χ_c_ [129], with pro-inflammatory cytokines increasing FGF receptors in pancreatic β-cells to enhance their survival [130]. As to whether this would indicate an FGF-2 driven increase in GSH and glutamate/AMPA-kainate-mGluR5/NAS-TrkB across pancreatic α- and β-cells will be important to determine. There is some evidence in other cell types to indicate that melatonin can increase FGF-2 [131], indicating that local melatonin derived from the mitochondrial melatonergic pathway may also act via FGF-2 induction. Pro-inflammatory cytokines, via IDO induction and kynurenine release, can activate the AhR to induce exhaustion in NK cells and CD8^+^ T cells, as occurs in the tumor microenvironment [38]. NK cells initially [132], and subsequently also CD8^+^ T cells, are the major drivers of the immune-mediated (autoimmune-like) destruction of pancreatic β-cells following the oxidative stress-induced MHC-1. The capacity of pancreatic β-cell derived kynurenine to activate the AhR and suppress NK cells and CD8^+^ T cells and wider autoimmunity [29] is highly likely to underpin the data showing the loss of pancreatic β-cell IDO to be one of the last events to occur before apoptosis [124]. As noted, kynurenine activation of the AhR in the cytoplasm and mitochondrial membrane of pancreatic β-cells leads to the ‘backward’ conversion of melatonin to NAS, which then can activate TrkB-T1 and or TrkB-FL to enhance pancreatic β-cell survival and insulin production [118]. As such, the capacity of pancreatic β-cells to drive the cytokine/IDO/kynurenine/AhR/NK cell/CD8^+^ T cell/NAS/TrkB pathway may be intimately linked to the suppression of autoimmune-like processes and the maintenance of pancreatic β-cell survival, whilst concurrently optimizing insulin production and pancreatic β-cell function. Clearly, the interface of the melatonergic pathway with pancreatic β-cell interactions with immune cells requires investigation. This could parallel recent thinking on the intercellular interactions occurring within the tumor microenvironment, defining T1DM as arising from interactions occurring within a pancreatic islet microenvironment. Section 6 highlights the relevance of wider intercellular interactions within pancreatic islets. 

One of the classical pathophysiological changes in T1DM is the build-up of amylin and its oligomerization, including in pancreatic β-cells. As with amyloid-β in dementia, amylin forms amyloid deposits that dysregulate homeostatic interactions among pancreatic islet cells. Recent work shows melatonin to prevent amylin oligomerization, as well as dissolving preformed fibrils [133]. A number of factors can upregulate amylin, including pro-inflammatory cytokines, with effects that involve upregulation of the transcription factors, NF-κB, and activator protein-1 (AP-1) [134]. It requires determination as to whether YY1 upregulates amylin in pancreatic β-cells, including via complex formation with AP-1, as shown in the promotors of other genes [135]. The capacity of the raised levels of NF-κB and YY1 to upregulate the melatonergic pathway would indicate that melatonin induction may be compromised in pancreatic β-cells, possibly as a consequence of the ‘backward’ conversion of melatonin to NAS, under conditions of pro-inflammatory cytokines driving the IDO/kynurenine/AhR pathway or from the suppression of the melatonergic pathway per se. The induction of both NAS and melatonin is dependent upon optimized mitochondrial function and the conversion of pyruvate to acetyl-CoA, indicating that one of the problems of suboptimal mitochondrial function is the suppressed capacity to upregulate the melatonergic pathway. This clearly requires investigation, given the protection afforded by melatonin against the oligomerization and damage caused by amylin in pancreatic β-cells, and may parallel recent work indicating the role of suppressed astrocyte melatonergic pathway in the accumulation of amyloid-β in dementia [97]. Beta-site amyloid precursor protein cleaving enzyme (BACE1) is also highly expressed in pancreatic β-cells where it acts to regulate insulin mRNA expression levels [136], whilst the inhibition of BACE2 increases pancreatic β-cell function and insulin production [137]. The interactions of the melatonergic pathway with BACE1 and BACE2 regulation and effects in pancreatic β-cells will be important to determine and would seem likely to parallel melatonin’s protection against amyloid-β production in the CNS [97]. 

Overall, it is clear from the direct effects of melatonin in pancreatic β-cells that the local regulation of the mitochondrial melatonergic pathway may be of some importance to a wide array of factors proposed to underpin pancreatic β-cell dysfunction and apoptosis in T1DM. 

## 6. Pancreatic Cells and Interactions

There is a growing appreciation that many complex conditions classically associated with dysregulation in a particular cell type may be better understood within an intercellular context. This is typified in cancers, where tumors shift macrophage metabolism to increase transforming growth factor (TGF)-β release that then suppresses NK cell and CD8^+^ T cell cytolytic responses [38]. The embracing of the complexity of such intercellular processes provides novel treatments that target intercellular processes. This poses the question as to whether the intercellular interactions of pancreatic and infiltrating cells provide a dynamic, intercellular microenvironment, the understanding of which may provide novel treatments. Dynamic, intercellular regulation of the mitochondrial melatonergic pathway is an important aspect of the tumor microenvironment that may provide some parallels to understanding the changes occurring in pancreatic β-cells in the course of T1DM. 

Approximately 98% of the pancreas is comprised of exocrine or acinar cells that secrete an array of digestive enzymes. Endocrine cells form the islets of Langerhans, being comprised of five different cell types, which are defined by the main hormone secreted into the bloodstream: namely, α-cells (glucagon), β-cells (insulin, amylin, C-peptide), δ-cells (somatostatin), ε-cells (ghrelin), and γ-cells (pancreatic polypeptide). α-cells comprise about 20–30% and β-cells 60–70% of these endocrine cells, which in the human pancreatic islets are scattered rather than clustered together, as in the murine islets [138]. Non-endocrine cells, including macrophages and endothelial cells, are also evident in the islets of Langerhans. 

Within the tumor microenvironment, it is proposed that the array of dynamic intercellular fluxes is determined by core processes regulating mitochondrial metabolism across different cell types, of which the mitochondrial melatonergic pathway is an important target. This allows tumors to form a new intercellular homeostasis that may be conceptualized as a form of evolutionary modified bacteria (mitochondria) interacting within, and across, cells [139]. This may be exemplified by the pro-inflammatory cytokine-induced increased IDO in pancreatic β-cells, with the release of kynurenine activating the AhR in an attempt to suppress the cytotoxicity of NK cells and CD8^+^ T cells by regulating their metabolism. This is an important intercellular process that allows cancers to not only survive the presence of NK cells and CD8^+^ T cells, but to also persuade these cytolytic cells via AhR activation, leading to the ‘backward’ conversion of melatonin to NAS, to potentially provide trophic support for cancer stem-like cell survival and proliferation via released NAS activating TrkB on cancer stem-like cells. Similar factors seem to be evident and important in T1DM, including the requirement for TrkB ligands (such as active muscle-derived BDNF) to optimize pancreatic β-cell function and insulin release [118]. The classical conceptualization of T1DM as being mediated solely by cellular changes in pancreatic β-cells may be avoiding the complexity of the dynamic homeostasis arising from intercellular metabolic interactions in which the mitochondrial melatonergic pathway is a significant factor. 

This poses more questions than answers from available data. Are pancreatic β-cells competitively eliminated within an altered intercellular homeostasis in islets? During brain development and in amyotrophic lateral sclerosis motor neuron loss, the induction of the truncated TrkB-T1 seems important to prevent BDNF and NAS trophic and metabolic support to cells [7]. Data in pancreatic β-cells do show BDNF, via TrkB-T1, to provide trophic support to pancreatic β-cells [118], although other data show that trophic support in pancreatic β-cells is provided mainly by TrkB-FL [122]. This requires independent investigation, including as to the role of alterations in mitochondrial metabolism in pancreatic β-cells as driven by a suppressed melatonergic pathway, leading to enhanced mitochondrial ROS and ROS-driven miRNAs, and therefore altered gene patterning. Different miRNAs can enhance (miR-34a, miR-4813) and decrease (miR-185) TrkB-T1 induction [140]. The relevance of the suppression of the mitochondrial melatonergic pathway to the differential regulation of TrkB-T1 regulating miRNAs will be important to determine. TrkB-T1 may be important in the proliferation of pancreatic β-cells in development [141], indicating that variations in mitochondrial metabolism and ROS production will shape the TrkB-FL/TrkB-T1 ratio, with concurrent consequences for patterned gene induction and intercellular communication. Whether there is a dynamic predominance of signaling via TrkB-FL vs. TrkB-T1 in pancreatic β-cells, determined by intercellular interactions within the pancreatic islet microenvironment, and driven by alterations in mitochondrial ROS-regulated miRNAs and associated changes in gene patterning will be important to determine. 

Whether there are primary or important alterations occurring in other pancreatic islet cells that changes the nature of the homeostatic interactions within the pancreatic islet microenvironment will be important to determine. For example, suboptimal mitochondrial function in α-cells, from suppressed system χ_c_, attenuates glutamate release and AMPA/kainate receptors activation, thereby forming the underpinnings of the lost glucagon release in T1DM [127]. The consequences of such changes in α-cells to the intercellular homeostatic regulation in the pancreatic islet microenvironment will be important to investigate. Wider processes in α-cells would also seem relevant, including the glutamate efflux in the course of GSH generation by system χ_c_ allowing α-cells antioxidant regulation to drive mGluR5 activation in pancreatic β-cells, thereby leading to the ‘backward’ conversion of melatonin to NAS and thereafter to the trophic and metabolic benefits of TrkB activation; see Figure 4. Intra-islet zinc homeostasis and zinc transporter 8 may also be an important variable in determining the homeostatic interactions among cells in the pancreatic islet [142]. The intercellular interactions of cells in the pancreatic islet microenvironment add a further layer of complexity to the pathoetiology of T1DM. 

## 7. Integrating T1DM Pathogenesis and Pathophysiology

The above highlights the diverse array of factors forming the biological underpinnings of T1DM. As an immune-mediated (classical autoimmune) disease, T1DM pathophysiology has significant overlaps with other immune-mediated diseases, including Parkinson’s disease. In Parkinson’s disease models, the induction of mitochondrial oxidative stress enhances MHC-1, which then increases the attraction of CD8^+^ T cells to the substantia nigra [143]. The mitochondrial melatonergic pathway is suppressed in the substantia nigra in Parkinson’s disease [67,144], indicating a relevant role for mitochondrial melatonergic pathway suppression in the pathophysiology of immune-mediated, classical ‘autoimmune’ disorders more widely. In Parkinson’s disease models, suppressed mitochondrial PTEN-induced kinase 1 (PINK1) is proposed to contribute to, if not underpin, MHC-1 upregulation [143], coupled to a decrease in mitophagy and associated metabolic dysregulation. Similar processes seem to occur in pancreatic β-cells, with the induction of oxidative stress, by a variety of means, decreasing mitophagy and PINK1, leading to metabolic dysregulation [145], in association with increased MHC-1, mitochondrial ROS-driven miRNAs, and altered gene patterning, thereby changing the nature of the interactions of pancreatic β-cells with other cells, including with NK cells and CD8^+^ T cells. 

These end-point changes overlap T1DM with many other conditions, including many that are not classically defined as autoimmune diseases, such as Parkinson’s disease. The primary alteration in such conditions is metabolic, thereby changing the interactions with other cells, including via alterations in the metabolism of proximal immune cells, given that immune cell activation requires glycolysis upregulation coupled to maintained OXPHOS [38,139]. The capacity to regulate the mitochondrial melatonergic pathway is crucial to the initiation and development of altered metabolic interactions of susceptible cells with immune cells. More specifically, the suppression of the mitochondrial melatonergic pathway in pancreatic β-cells attenuates melatonin induction of PINK1 on the mitochondrial membrane and its interactions with Parkin and leucine zipper-EF hand-containing transmembrane protein 1 (LETM1), which underpin mitophagy.

LETM1 localizes to the mitochondrial membrane, where it has an array of functions, including the maintenance of mitochondrial shape and cell viability. LETM1 is a putative Ca^2+^/H^+^ antiporter that regulates autophagy and mitochondrial oxidative stress [146]. LETM1 function is upregulated following phosphorylation by PINK1, suggesting that mitochondrial Ca^2+^ dysregulation may be linked to processes underpinning oxidative stress and MHC-1 upregulation. In pancreatic β-cells, LETM1 activation, as with the mitochondrial Ca^2+^ uniporter (MCU), regulates Ca^2+^-induced matrix acidification, providing a nutrient-induced mitochondrial pH gradient, which the authors claim is crucial for maintained ATP synthesis and the coupling of metabolism with pancreatic β-cell secretion.

Interestingly, the mitochondrial matrix tail of LETM1 has a 14-3-3-like motif, which may have the capacity to bind 14-3-3 and/or AANAT, thereby being intimately linked to mitochondrial melatonergic pathway regulation, whilst being in close proximity to mitochondrial ribosomes [147]. Importantly, exogenous melatonin promotes PINK1 accumulation on the mitochondrial membrane, as shown in neurons [148], indicating that melatonin not only upregulates the beneficial effects of PINK1 regarding autophagy, mitophagy, and protection against oxidative stress, but would also prevent the consequences of suppressed PINK1, including Ca^2+^ dysregulation, oxidative stress, and associated MHC-1 upregulation [149]. The interactions of LETM1, PINK1, Parkin, AANAT, and 14-3-3 will be important to determine in pancreatic β-cells, including as to the consequences arising from the upstream suppression of the melatonergic pathway; see Figure 5. 

Many of the factors reviewed above impact on T1DM pathophysiology via their modulation of mitochondrial function, including gut microbiome-derived butyrate, which increases mitochondria function via the upregulation of mitochondria-located sirtuin-3 and the deacetylation and disinhibition of PDC, leading to the conversion of pyruvate to acetyl-CoA, and thereby to increased ATP from the TCA cycle and OXPHOS. Gut permeability, via LPS activation of TLR4/NF-κB-YY1, drives inflammatory processes, which are time-limited by the temporal synchronization of NF-κB and YY1 with the induction of the mitochondrial melatonergic pathway. Many of the T1DM susceptibility genes act to regulate MHC-1 and mitochondrial function [17,22], highlighting the importance of mitochondrial metabolism and how it interfaces with classical ‘autoimmune’ associated processes. Clearly, optimized mitochondrial ATP production is crucial to how nutrient-stimulated K_ATP_ channel closure/opening regulates insulin secretion, providing a ready metabolic link to the role of K_ATP_ channels in pancreatic insulin secretion [94]. The differential expression of mitochondria-targeting miRNAs in T1DM vs. Controls highlights the importance of alterations in mitochondrial metabolism in T1DM [96]. RAGE, HMGB1, the AhR, and circadian dysregulation may all act to dysregulate mitochondrial metabolism and the mitochondrial melatonergic pathway, with consequences that include alterations in the dynamic, intercellular fluxes between cells. 

The dependence of pancreatic β-cells on TrkB-FL [122] and TrkB-T1 [118] highlights the importance of BDNF signaling, including from muscle activity, in the maintenance of optimized pancreatic β-cell function and insulin release. These data also strongly implicate variations in the melatonergic pathway, specifically the factors regulating the NAS/melatonin ratio, including kynurenine at the AhR, ATP at the P2Y1r, and glutamate at the mGluR5. Such data indicate the relevance of variations in the mitochondrial melatonergic pathway in pancreatic β-cell plasticity, and how the suppression of the melatonergic pathway can have metabolic and intercellular consequences for pancreatic β-cells, including in driving immune-mediated ‘autoimmune’-like changes in immune responses. It will be important to determine whether mitochondrial ROS, and its regulation by the mitochondrial melatonergic pathway, impacts on TrkB-T1 regulating miRNAs, namely miR-34a, miR-185, and miR-4813, and therefore on the consequences of BDNF- and NAS-driven signaling. 

The beneficial effects of *Lactobacillus johnsonii* strain N.6 and released nanovesicles in T1DM will be important to determine. Data indicate that fungal infection with *Candida albicans* in the gut is a significant factor in T1DM pathoetiology [85]. As *Lactobacillus johnsonii* can eliminate *Candida albicans* from the gut [86], the application of *Lactobacillus johnsonii* may prove a significant treatment advance in the prevention of pancreatic β-cell loss at first presentation. Although the proliferation of *Candida albicans* may be arising from suppression of NK cell and CD8^+^ T cell responses, it will be interesting to determine how *Candida albicans* can drive alterations in pancreatic β-cell mitochondrial metabolism. The most direct route would seem via alterations in gut microbiome-derived products, such as butyrate, although this requires future investigation. 

Although requiring further investigation, it would seem clear that alterations in pancreatic β-cells will involve changes in their interactions with other cells in the pancreatic islets. As exemplified above in α-cells, changes in other cell types within islets may be significant, and possibly primary, via their interactions with pancreatic β-cells. The inability to resolve inflammation locally leads to circulating pro-inflammatory cytokines that suppress pineal melatonin production and increase gut permeability and associated gut dysbiosis [121]. This limits the capacity of night-time pineal melatonin and gut microbiome-derived butyrate to re-establish homeostasis via their capacity to optimize mitochondrial function across different cells. Both pineal melatonin and butyrate act, at least in part, via the upregulation of the mitochondrial melatonergic pathway. However, if this is suppressed in pancreatic β-cells the ability of melatonin and butyrate to achieve this will be curtailed. 

The above highlights processes that form the pathophysiology of T1DM and the importance of factors dysregulating the mitochondrial melatonergic pathway in pancreatic β-cells. However, the pathoetiology of T1DM is still the subject of intense investigation. Recent work has highlighted alterations in the wider gut microbiome, including significant roles for enteroviruses and bacteriophages in the pathogenesis and pathophysiology of T1DM [77,78], ultimately resulting in changes in gut bacteria and suppression of the short-chain fatty acids, such as butyrate. As noted, recent work has indicated pathophysiological overlaps of amyotrophic lateral sclerosis (ALS) and T1DM [4], with ALS showing significant changes in the gut microbiome, including as induced by glyphosate-based herbicides (GBH), which suppress the shikimate pathway [7]. In humans and animals, the shikimate pathway is primarily achieved by *Akkermansia muciniphila* [150]. In poultry, *Akkermansia muciniphila* is significantly suppressed by GBH, and does not seem to be restored following GBH cessation [151]. Preclinical data show GBH to suppress butyrate and propionate [151] and to bias the thriving of some bacteria, as indicated by an increase in α-diversity [152]. Interestingly, *Akkermansia muciniphila* is a significant regulator of T1DM pathoetiology [153], with the transfer of *Akkermansia muciniphila* remodeling the gut microbiome, maintaining the gut barrier, reducing circulating LPS and TLR expression, as well as reducing pancreatic islet autoimmunity and significantly delaying T1DM onset in non-obese diabetic (NOD) mice [154]. In T1DM patients, *Akkermansia muciniphila* levels negatively correlate with glucose level and HbA1c [155], indicating a significant association with T1DM pathophysiology. Probiotic administration, including of *Lactobacillus johnsonii*, increases *Akkermansia muciniphila* in T1DM patients, in association with improved glucose control and HbA1c levels [156]. Interactions of *Akkermansia muciniphila* with bacteriophages can regulate *Akkermansia muciniphila* levels [157], indicating important bacteriophage impacts via *Akkermansia muciniphila* and the shikimate pathway.

The suppression of the shikimate pathway decreases not only tryptophan but also tryptophan-derived ligands for the AhR, such as tryptamine and indole-3-acetate, thereby indicating an association of a suppressed shikimate pathway not only with gut permeability, but also with the regulation of NK cell and CD8^+^ T cell cytotoxicity [38]. The suppressed levels of *Akkermansia muciniphila* in T1DM may therefore be intimately linked to lower shikimate pathway-derived tryptophan (as well as tyrosine and phenylalanine), increased gut permeability, and an altered capacity of the gut microbiome to suppress NK cells and CD8^+^ T cells. Although AhR activation by kynurenine can induce a state of ‘exhaustion’ in NK cells and CD8^+^ T cells [38], the AhR has differential effects in memory CD8^+^ T cells, with AhR activation suppressing circulating memory CD8^+^ T cells, while promoting the core gene program of resident memory CD8^+^ T cells [158]. As the gut is proposed to be an important site for the inappropriate maintenance of autoreactive memory CD8^+^ T cells, whereby autoimmune-linked autoreactive CD8^+^ T cells may interact in the Peyer’s patches of the gut to escape thymic deselection [159], the effects of enteroviruses and bacteriophages in T1DM pathoetiology may be mediated via suppressed *Akkermansia muciniphila* levels and shikimate pathway activity. The potentiation of effector functions of autoreactive T cells in the gut seems to be via bystander activation and not from “molecular mimicry”, as could arise from cross-reactivity between gut microbiota-derived peptides and islet-derived epitopes [159]. In a preclinical T1DM model, these authors show that the initial activation of islet specific CD8^+^ T cells occurs in the pancreatic lymph nodes but with additional effector function occurring in the gut lymphoid tissues via non-specific bystander activation [159]. These authors also showed that the oral administration of the short-chain fatty acid, butyrate, attenuated the ‘bystander’ induced cytotoxic effector functions of these autoreactive CD8^+^ T cells [159].

This requires further investigation as it would indicate that initial changes occurring in the intercellular interactions of pancreatic islet cells drive alterations in the mitochondrial melatonergic pathway in pancreatic β-cells that decreases PINK1/parkin/LETM1/mitophagy and increases MHC-1 that primes CD8^+^ T cells that would normally be subject to thymic deselection. However, the changes in the wider gut microbiome, including as induced by bacteriophages, enteroviruses, and possibly GBH, suppress *Akkermansia muciniphila* and the shikimate pathway to provide a gut microenvironment, possibly in Peyer’s patches and involving variations in different AhR ligands and levels, which allows these autoreactive CD8^+^ T cells to escape thymic deselection. This would indicate that alterations in the gut microbiome are relevant to different aspects of T1DM pathoetiology and pathophysiology, namely changes in intercellular interactions of cells in pancreatic islets as well as in the maintenance of memory CD8^+^ T cells arising from CD8^+^ T cells initially activated due to the intercellular interactions driving changes in the mitochondrial melatonergic pathway of pancreatic β-cells. 

The data reviewed above have a number of research and treatment implications.

## 8. Future Research

Is the melatonergic pathway evident in pancreatic β-cells? Is the pancreatic β-cell melatonergic pathway suppressed in T1DM?Is the suppression of the mitochondrial melatonergic pathway relevant in diverse tissues and organs associated with T1DM comorbidities, either directly suppressed in these organs/tissues or via alterations in patterned immune responses?How does the suppression of the mitochondrial melatonergic pathway link to, or drive, immune-mediated processes underpinning classical concepts of autoimmunity? Is a suppressed mitochondrial melatonergic pathway relevant in other immune-mediated, autoimmune disorders, redefining these conditions as a subtype of mitochondrial disorders?Are there parallels to the tumor microenvironment, where other cells (as with cancer cells) act to suppress the mitochondrial melatonergic pathway in pancreatic β-cells, thereby allowing local intercellular processes to drive ‘autoimmunity’ via miRNA-induced MHC-1?What is the relative importance of the TrkB-FL vs. TrkB-T1 in pancreatic β-cells? Does NAS, vs. BDNF, have any differential effects at TrkB-FL vs. TrkB-T1 in pancreatic β-cells? Is the TrkB-FL/TrkB-T1 ratio determined by mitochondrial ROS driving miRNA patterning and/or in association with the NAS/melatonin ratio?Skin symptoms are evident in T1DM, with streptozotocin altering skin function, including decreasing fibroblast growth factors (FGFs), especially during wound healing [160]. As melatonin can increase FGFs [161], are the effects of streptozotocin in the skin driven by streptozotocin (and T1DM?) suppression of the tryptophan-melatonin pathway in skin cells?As human amylin increases TLR-4 in rodents [162], does this indicate an amylin-driven increase in the TLR-4/NF-κB-YY1 induction of inflammatory changes in pancreatic β-cells, with the effects of amylin dependent upon the capacity of NF-κB and YY1 to upregulate the melatonergic pathway?Given that amylin is normally secreted with insulin, would YY1, like NF-κB [134], upregulate amylin in pancreatic β-cells?Does gut microbiome-derived butyrate, including as regulated by *Lactobacillus johnsonii* and its suppression of *Candida albicans*, have a role in the pathoetiology and pathophysiology of T1DM? Do the effects of butyrate in pancreatic β-cells require a functional tryptophan-melatonin pathway?Does TPH1 require stabilization by a specific 14-3-3 isoform, as with TPH2 by 14-3-3eta, to convert tryptophan to serotonin in pancreatic β-cells?Prenatal serotonin effects in pancreatic β-cells have been proposed to upregulate adult pancreatic β-cell numbers [163], although this is not supported by data using TPH1 knockout [164]. Is there a relevant role for the tryptophan–melatonin pathway in pancreatic β-cells during development? Would the survival and proliferative/functional importance be lost when no neighboring cells express the melatonergic pathway, indicating the importance of the tryptophan–melatonin pathway to challenging, if not competitive, intercellular interactions?In other cell types, exogenous melatonin (local, autocrine, paracrine, circadian) can be taken up into mitochondria via the organic anion transporter (OAT)3 and the peptide transporters (PEPT)1/2 [165]. OAT is expressed in pancreatic β-cells, and so whether OAT3 and/or PEPT1/2 are present in the mitochondrial membrane in pancreatic β-cells will be important to determine.T1DM is associated with an increased risk of amyotrophic lateral sclerosis (ALS) in people aged < 50 years of age [166], with streptozotocin-induced T1DM also leading to neuromuscular junction retraction and muscle atrophy [167]. There also seems genome-wide genetic overlaps between T1DM and ALS [168], whilst glyphosate-based herbicides, a proposed risk factor for ALS [7], induces a T2DM phenotype when combined with a high fructose diet [169]. Chronic glyphosate causes severe degeneration in pancreatic acinar cells and islets of Langerhans [170]. This could suggest epigenetic and genetic overlaps of ALS and T1DM. Is this mediated via gut, immune, and/or mitochondrial melatonergic-related factors? Are glyphosate-based herbicides an environmental risk factor for T1DM? Glyphosate-based herbicides can inhibit the shikimate pathway, which is a relevant provider of tryptophan to the body, with the inhibition of the shikimate pathway increasing gut permeability and gut dysbiosis, including decreased butyrate producing gut bacteria [7]. Do enteroviruses and/or bacteriophages in T1DM suppress the shikimate pathway and *Akkermansia muciniphila*, in association with decreased *L. johnsonii* and butyrate, to contribute not only to alterations in the mitochondrial melatonergic pathway in pancreatic β-cells, but also to the ‘bystander activation’ of memory CD8^+^ T cells in Peyer’s patches? Do such alterations in the gut enhance the cytotoxicity of autoreactive CD8^+^ T cells and prevent their elimination/deselection in the thymus?How relevant is the modulation of PINK1 by melatonin to the interactions of LETM1, PINK1, and Parkin in the mitochondrial membrane? Does the 14-3-3-like domain of LETM1 bind 14-3-3 and/or AANAT to regulate the mitochondrial melatonergic pathway? Would this more directly link autophagy with mitochondrial melatonergic pathway regulation?By limiting the oxidative stress-induced DNA damage, and therefore the induction of PARP1, does melatonin increase the availability of NAD^+^ for sirtuin induction, thereby increasing PDC and mitochondrial OXPHOS [171]?Alterations in tryptophan and serotonin are evident in gestational diabetes [172], indicating consequences for the mother, placenta, and offspring [173], including in the regulation of the mitochondrial melatonergic pathway. This will be important to clarify in future research.

## 9. Treatment Implications

Preclinical data show that the administration of soluble RAGE dramatically suppresses T1DM via Treg upregulation, and associated inhibition of conventional T cell division [27]. Such data would implicate a significant clinical impact of RAGE ligands that can be suppressed by soluble RAGE. As melatonin inhibits RAGE ligands and RAGE activation in diabetes models [28], this could implicate the utilization of melatonin in the suppression of RAGE-driven T1DM pathophysiology.Would *Lactobacillus johnsonii* prove useful in suppressing progressive pancreatic β-cell loss at initial T1DM diagnosis? Is the efficacy of *Lactobacillus johnsonii* only evident when *Candida albicans* is present in the gut?The green tea polyphenol, epigallocatechin gallate (EGCG), binds monomeric amylin, preventing its aggregation [174], indicating that it may have similar utility to recently developed small molecule inhibitors of amylin fibrillation [175].An array of different pharmaceuticals and nutriceuticals afford protection in pancreatic β-cells via the suppression of the NLRP3 inflammasome [176,177,178]. Although not investigated in pancreatic β-cells, melatonin suppresses the NLRP3 inflammasome in different cell types across the body [179], indicating another aspect to melatonin’s potential utility in T1DM treatment, as well as from locally produced pancreatic melatonin.Stem cell development now allows for the priming of stem cells to have the contents of their exosome/vesicles shaped to provide targeted treatments (such as miRNAs and 14-3-3 proteins) to particular cells. This requires the identification of relevant targets. Would targeting the melatonergic pathway in pancreatic β-cells optimize mitochondrial function, whilst decreasing oxidative stress-induced MHC-1 and there preventing NK cell and CD8^+^ T cell induced apoptosis?A peptide-based therapy, JC-1 ScFv, has recently been found to bind specifically to the *Candida albicans* cell wall, where it inhibits the growth and viability of *Candida albicans*, both in vitro and in vivo [180]. As well as the utilization of *Lactobacillus johnsonii* for the management of *Candida albicans* in T1DM, such peptide-based therapies may provide another treatment option.A plethora of preclinical studies indicate the utility of melatonin in attenuating many of the consequences of T1DM, including the following: cardiovascular disorders [181], pancreatic β-cell regeneration [109], renal impairment [182], bone loss [183], cognitive deficits [110], erectile dysfunction [184], and body temperature circadian rhythm [185]. This is supported by clinical data showing lower melatonin levels in T1DM children [186]. The capacity of melatonin to decrease gut dysbiosis and gut permeability, as well optimize mitochondrial function, will be relevant to the management of many aspects of T1DM pathophysiology.

## 10. Conclusions

The inclusion of the mitochondrial melatonergic pathway in pancreatic β-cells and the cells with which β-cells interact can integrate wide bodies of previously disparate data on T1DM. Mitochondrial dysfunction is clearly the major pathophysiological change occurring in pancreatic β-cells in T1DM. The protection afforded by mitochondrial melatonin is likely to include the optimization of mitophagy as well as the suppression of oxidative stress, and therefore the suppression of ROS-driven miRNAs that alter gene patterning. The oxidative stress dysregulation of PINK1 and mitophagy seems to be coupled to raised MHC-1 levels, thereby driving immune-mediated pancreatic β-cell destruction. The relevance of the melatonergic pathway is further emphasized by the immediate precursor of melatonin, namely NAS, being a BDNF mimic via TrkB activation. Given the important role of both TrkB-FL and TrkB-T1 in pancreatic β-cell function and survival, the regulation of the local ligand for this receptor clearly requires investigation, including as to how alterations in the gut microbiome can interface with the mitochondrial melatonergic pathway in pancreatic β-cells and local interacting cells. The suppression of *A. muciniphila* and the shikimate pathway, as well as butyrate and *L. johnsonii*, not only contributes to alterations in pancreatic cell intercellular interactions and suppressed mitochondrial melatonergic pathway activation in pancreatic β-cells but also provides a gut microenvironment, possibly in Peyer’s patches, whereby autoreactive CD8^+^ T cells acquire enhanced effector functions and avoid deselection in the thymus. This provides a framework to target T1DM pathophysiology and not only T1DM symptoms. 

## Figures and Tables

**Figure 1 ijms-24-03300-f001:**
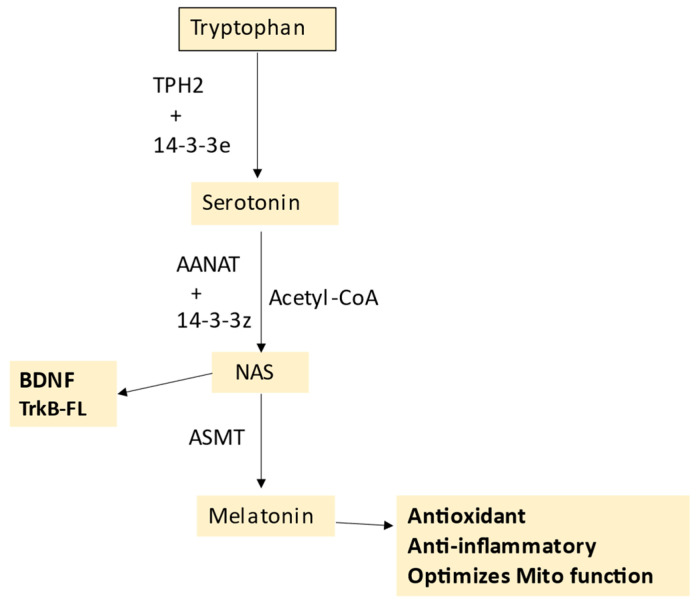
The tryptophan–melatonin pathway. Tryptophan is converted to serotonin by tryptophan hydroxylase (TPH)1 or TPH2, which requires stabilization by 14-3-3e. AANAT converts serotonin to *N*-acetylserotonin (NAS), with AANAT requiring stabilization by another 14-3-3 isoform and the presence of acetyl-CoA as a co-substrate. ASMT converts NAS to melatonin. NAS is a BDNF mimic, via its activation of the BDNF receptor, TrkB. NAS may also induce BDNF. Melatonin is a powerful antioxidant and anti-inflammatory that optimizes mitochondrial function. Abbreviations: AANAT: aralkylamine *N*-acetyltransferase; Acetyl-CoA: acetyl coenzyme A; ASMT: *N*-acetylserotonin *O*-methyltransferase; BDNF: brain-derived neurotrophic factor; NAS: *N*-acetylserotonin; TPH: tryptophan hydroxylase.

**Figure 2 ijms-24-03300-f002:**
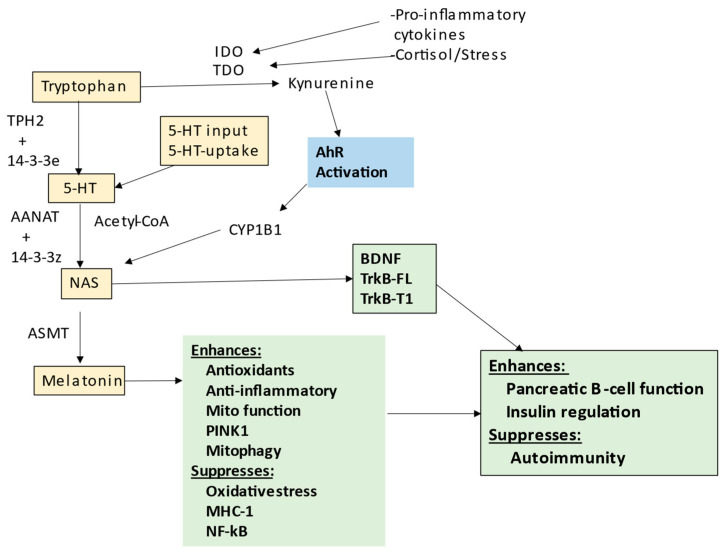
Shows the tryptophan–melatonin pathway (gold shade). Tryptophan is converted by tryptophan hydroxylase (TPH2 stabilized by 14-3-3e) to serotonin (5-HT), which is the necessary precursor for the melatonergic pathway. 5-HT can also be provided by neuronal inputs and other cellular sources, including platelets. In the presence of acetyl-CoA, 5-HT is converted by 14-3-3 stabilized AANAT to *N*-acetylserotonin (NAS), which is then converted to melatonin by AANAT. Under inflammatory conditions, as in T1DM, cytokines increase indoleamine 2,3-dioxygenase (IDO) and TDO, which converts tryptophan to kynurenine, suppressing tryptophan levels. Kynurenine also activates the aryl hydrocarbon receptor (AhR), which can increase the NAS/melatonin ratio, as well as suppress available melatonin. NAS increases BDNF and can activate the TrkB receptors. Melatonin has many protective effects as well as suppressing oxidative stress and MHC-1 linked autoimmunity, including in pancreatic B-cells. Abbreviations: 5-HT: serotonin; AANAT: aralkylamine *N*-acetyltransferase; AhR: aryl hydrocarbon receptor; ASMT: *N*-acetylserotonin *O*-methyltransferase; CYP: cytochrome P450; IDO: indoleamine 2,3-dioxygenase; MHC-1 major histocompatibility complex-class 1; NAS: *N*-acetylserotonin; NF-κB: nuclear factor kappa-light-chain-enhancer of activated B cells; PINK1: PTEN-induced kinase 1; TDO: tryptophan 2,3-dioxygenase; TrkB-FL: tyrosine receptor kinase B-full length; TrkB-T1: tyrosine receptor kinase B-truncated.

**Figure 3 ijms-24-03300-f003:**
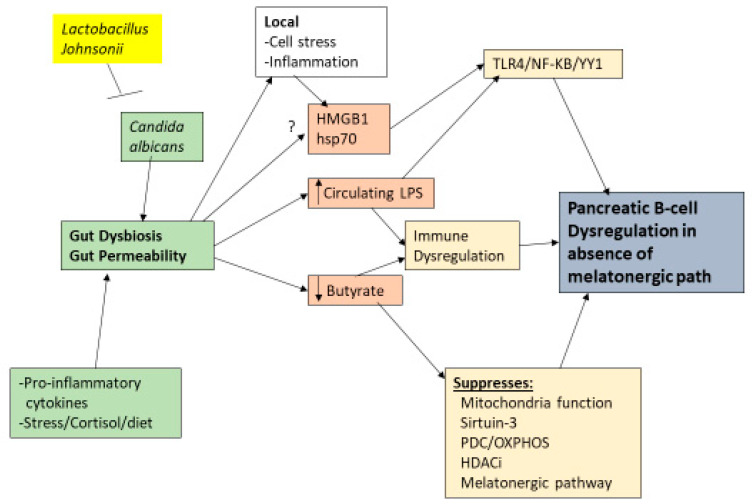
Shows the role of gut dysbiosis and gut permeability in T1DM. *Candida albicans* fungal infection in the gut can lead to gut dysbiosis and associated gut permeability, as can heightened pro-inflammatory cytokines, stress/cortisol, and many dietary factors. *Lactobacillus johnsonii* bacteria can eliminate *Candida albicans* from the gut. Gut permeability increases circulating LPS, which can dysregulate the immune response via TLR4 activation, leading to pro-inflammatory transcription factors, NF-κB and YY1, in immune cells and pancreatic B-cells. Butyrate suppression attenuates its optimization of mitochondrial function via increased sirtuin-3 and PDC induction that enhances OXPHOS, at least partly via the melatonergic pathway. Suppressed butyrate will also attenuate its capacity as a HDACi, leading to altered epigenetic regulation, with consequences for local cellular stress and inflammation, which increases HMGB1 and hsp70. Both of these TLR4 ligands may also be released by the gut, including in exosomes. These gut-derived changes will have direct impacts on pancreatic B-cells and other pancreatic islet cells as well as indirect effects via alterations in patterned immune responses. Damage in pancreatic B-cells will be at least partly dependent upon the suppression of the melatonergic pathway. Abbreviations: HDACi: histone deacetylase inhibitor; HMGB1: high-mobility group box 1; hsp70: heat shock protein 70; LPS: lipopolysaccharide; OXPHOS: oxidative phosphorylation; NF-κB: nuclear factor kappa-light-chain-enhancer of activated B cells; PDC: pyruvate dehydrogenase complex.

**Figure 4 ijms-24-03300-f004:**
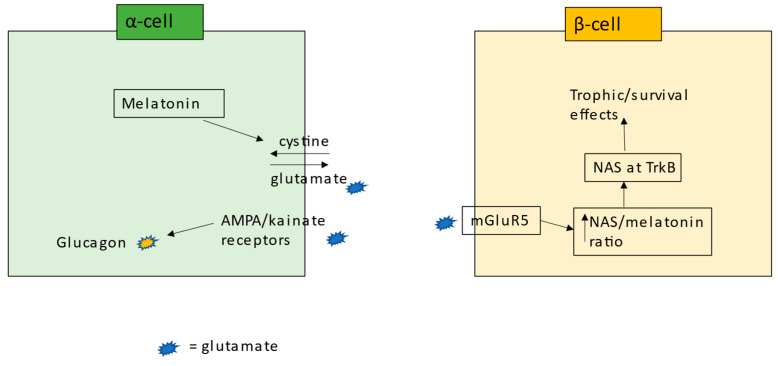
Shows how alterations in pancreatic α-cells, including melatonin production and its upregulation of the cystine-glutamate antiporter (system χ_c_), may act to upregulate NAS in pancreatic β-cells via mGluR5 activation, thereby increasing trophic and survival processes in pancreatic β-cells. Abbreviations: AMPA: alpha-amino-3-hydroxy-5-methyl-4-isoxazolepropionic acid; mGluR5: metabotropic glutamate receptor 5; NAS: *N*-acetylserotonin; TrkB: tyrosine kinase receptor B.

**Figure 5 ijms-24-03300-f005:**
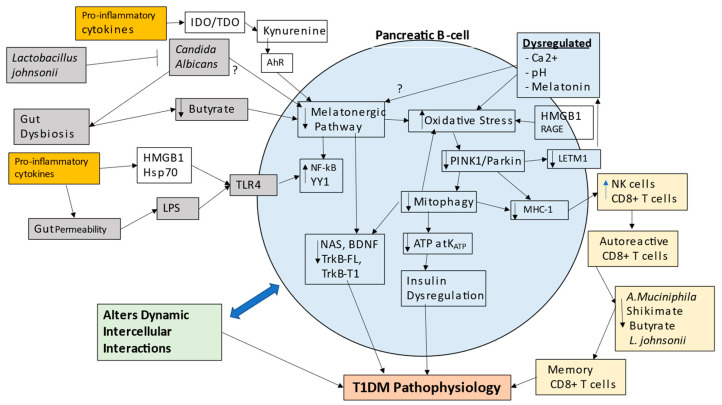
Shows how gut dysbiosis, gut permeability, pro-inflammatory cytokines, and *Candida albicans* fungal infection act to suppress the mitochondrial melatonergic pathway in pancreatic β-cells. The suppressed capacity to upregulate melatonin prolongs the heightened activation of pro-inflammatory signaling via the transcription factors, NF-κB and YY1, coupled to decreased activation of TrkB-FL and/or TrkB-T1 by NAS and BDNF. A suppressed mitochondrial melatonergic pathway enhances oxidative stress, thereby decreasing PINK1 and its interactions with parkin and LETM1 on the mitochondrial membrane. Decreased PINK1 suppresses mitophagy, coupled to increased MHC-1 that drives ‘autoimmune’ processes via NK cell and CD8^+^ T cell attraction. The accompanying decrease in OXPHOS-derived ATP prevent K_ATP_ induced insulin, whilst decreased PINK1 attenuates LETM1 phosphorylation, leading to Ca^2+^ and pH dysregulation, likely accompanied by alterations in how LETM1 interacts with 14-3-3 and/or AANAT in the regulation of the mitochondrial melatonergic pathway. As well as activating TLR4, HMGB1 activates RAGE, thereby further contributing to oxidative stress. Changes in pancreatic β-cell mitochondrial function, including by ROS-driven miRNAs, will change patterned gene induction, with consequent changes in fluxes that mediate pancreatic β-cell interactions with other cells in the pancreatic islet microenvironment, thereby changing the dynamic intercellular interactions occurring. The decrease in shikimate pathway, *A. muciniphila*, *L. johnsonii*, and butyrate, contributed to by bacteriophages and enteroviruses, provides ‘bystander’ activation of autoreactive CD8^+^ T cells—possibly in Peyer’s patches—thereby preventing thymic deselection and driving classical ‘autoimmunity’. Abbreviations: AhR: aryl hydrocarbon receptor; BDNF: brain-derived neurotrophic factor; HMGB: high-mobility group box; hsp: heat shock protein; IDO: indoleamine 2,3-dioxygenase; K_ATP_: ATP-activated potassium channel; LETM1: leucine zipper-EF hand-containing transmembrane protein 1; LPS: lipopolysaccharide; MHC-1: major histocompatibility complex-class 1; NAS: *N*-acetylserotonin; NF-κB: nuclear factor kappa-light-chain-enhancer of activated B cells; RAGE: receptor for advanced glycation end-products; NK: natural killer; TDO: tryptophan 2,3-dioxygenase; TrkB-FL: tyrosine kinase receptor B-full length; TrkB-T1: tyrosine kinase receptor B-truncated; YY1: yin yang 1.

## Data Availability

Not applicable.

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
