# Peer review of "Type I Diabetes Pathoetiology and Pathophysiology: Roles of the Gut Microbiome, Pancreatic Cellular Interactions, and the ‘Bystander’ Activation of Memory CD8+ T Cells"

_ijms, 2023, doi:10.3390/ijms24043300_

Round 1

Reviewer 1 Report

This is an interesting article in which the author reviewed the complexity of type I diabetes pathoetiology with emphasis in alterations of the mitochondrial melatonergic pathway of pancreatic B-cells. Considering the amount of interacting molecules and pathways described in this review (including microbiome and miRNAs), naming only three molecules in the title (YY1, NF-kB and TLR4) seems ‘reductionist’, I suggest rewriting the title to better represent everything discussed in this review.

In addition, the amount of information presented in this review is sometimes difficult to follow and would be better understood with the help of some figures; for example a figure integrating signaling pathways in TIDM pathophysiology, and/or a figure representing mitochondrial function in pancreatic β-cells, and/or a figure representing pancreatic islet microenvironment and cell interactions. I strongly suggest a figure integrating T1DM pathogenesis and pathophysiology (section 7) and/or a graphical abstract.

In section 2, the author discusses RAGE activation and administration of soluble RAGE, mentioning “Preclinical data shows that the administration of soluble RAGE dramatically suppresses T1DM via Treg upregulation, and associated inhibition of conventional T cell division. Such data would implicate a significant clinical impact of RAGE ligands that can be suppressed by soluble RAGE and melatonin...” however, at this point melatonin has not been introduced to the reader, therefore it should not be mentioned (yet) or it should be introduced.

In section 3.2, the author discussed the gut microbiome in relation to T1DM, however maternal endometrial and/or vaginal microbiomes of diabetic mothers (including GDM) may increase (through epigenetic mechanisms) the risk of the offspring to develop T1DM (as it has been documented for T2DM). Please discuss.

Consistency of terminology (T1DM or T1D) should be revised throughout the document. Also I suggest writing “T cells” instead of “t cells”.

Be aware of nomenclature for scientific names: Lactobacillus johnsonii, and Candida albicans (italics, genus starts with capital letter) throughout the document and in Figure 2.

There are several minor mistakes and typos. Here I indicate some of them but there may be more. Please revise the entire document. It would be helpful including line numbers. 

Section 2, 3rd line, “…pancreatic B-cells remain open…”

Section 3.1 7th line, I believe it should say “…by endogenous and exogenous ligands…”; line 12: “…cells and tissues…”; 4th paragraph “YY1 ko” what does ko stand for? Or it may be a typo?; 5th paragraph: …”would indicate…”

Section 3.2: “…which may be contributed to by alterations in the gut microbiome.” Please correct the sentence.

Figure 1: change “Osidative stress” for “Oxidative stress”

Author Response

Journal: IJMS (ISSN 1422-0067)

Manuscript ID: ijms-2168614       Type: Review

Title:  Type I diabetes pathoetiology and pathophysiology: Role of suppressed Akkermansia muciniphila, Lactobacillus johnsonii, shikimate pathway, and butyrate in the gut microbiome in driving dysregulated intercellular interactions in the pancreas and the ‘bystander’ activation of memory CD8+ T cells.

Author: George Anderson      Section: Molecular Endocrinology and Metabolism

Special Issue: Central and Peripheral Molecular Mechanisms of Metabolism Regulation 2.0

Response to Reviewers

Reviewer 1

This is an interesting article in which the author reviewed the complexity of type I diabetes pathoetiology with emphasis in alterations of the mitochondrial melatonergic pathway of pancreatic B-cells.

Response to Reviewer

Thank you for these encouraging comments.

-Considering the amount of interacting molecules and pathways described in this review (including microbiome and miRNAs), naming only three molecules in the title (YY1, NF-kB and TLR4) seems ‘reductionist’, I suggest rewriting the title to better represent everything discussed in this review.

Response to Reviewer

Thank you for highlighting this. The title has now been changed to:Type I diabetes pathoetiology and pathophysiology: Role of suppressed Akkermansia muciniphila, Lactobacillus johnsonii, shikimate pathway, and butyrate in the gut microbiome in driving dysregulated intercellular interactions in the pancreas and the ‘bystander’ activation of memory CD8+ T cells.

-In addition, the amount of information presented in this review is sometimes difficult to follow and would be better understood with the help of some figures; for example a figure integrating signaling pathways in TIDM pathophysiology, and/or a figure representing mitochondrial function in pancreatic β-cells, and/or a figure representing pancreatic islet microenvironment and cell interactions. I strongly suggest a figure integrating T1DM pathogenesis and pathophysiology (section 7) and/or a graphical abstract.

Response to Reviewer

 I agree. A summary figure (figure 5) has now been added to integrate the complexity of processes highlighted in section 7.  A figure (figure 1) introducing the melatonergic pathway is also added to Section 2, Another new figure (Figure 4) to highlight how local interactions may be occurring within the pancreatic islets. Figures 2 and 3 have been maintained, giving a total of 5 figures.

In section 2, the author discusses RAGE activation and administration of soluble RAGE, mentioning “Preclinical data shows that the administration of soluble RAGE dramatically suppresses T1DM via Treg upregulation, and associated inhibition of conventional T cell division. Such data would implicate a significant clinical impact of RAGE ligands that can be suppressed by soluble RAGE and melatonin...” however, at this point melatonin has not been introduced to the reader, therefore it should not be mentioned (yet) or it should be introduced.

 Response to Reviewer

As Section 2 deals with classical T1DM pathophysiology, I don’t think introducing data on melatonin would fit well at this point. However, I agree that a brief mention of melatonin effects would be good at this point in the manuscript and have added: “The antioxidant, anti-inflammatory, circadian and mitochondrial optimizing effects of melatonin are important aspects of T1DM, whilst the immediate precursor of melatonin, N-acetylserotonin (NAS) may also be a crucial regulator of pancreatic β-cell survival, via its capacity to activate the brain-derived neurotrophic factor (BDNF) receptor TrkB, as shown in figure 1. The role of the melatonergic pathway in T1DM is detailed in section 5.”  A figure and legend describing the tryptophan-melatonin pathway is also provided at this point.

In section 3.2, the author discussed the gut microbiome in relation to T1DM, however maternal endometrial and/or vaginal microbiomes of diabetic mothers (including GDM) may increase (through epigenetic mechanisms) the risk of the offspring to develop T1DM (as it has been documented for T2DM). Please discuss.

 Response to Reviewer

Thank you for highlighting this general point on microbiota. The following has now been added to Section 3.2: “The role of microbiota in T1DM and T2DM is further highlighted by data showing the impact of the maternal endometrial and/or vaginal microbiomes of diabetic mothers, including in gestational diabetes mellitus, which can act through epigenetic mechanisms to increase T2DM, and possibly T1DM, risk in the offspring [67]. Such data highlights the importance of microbiota in the regulation of metabolism, and the likelihood of prenatal priming of later T1DM related processes in pancreatic β-cells. This would require future investigation.”

The following reference has also been added:

Bankole, T.; Winn, H.; Li, Y. Dietary Impacts on Gestational Diabetes: Connection between Gut Microbiome and Epigenetic Mechanisms. Nutrients. 2022, 14(24), 5269. doi: 10.3390/nu14245269.

Consistency of terminology (T1DM or T1D) should be revised throughout the document. Also I suggest writing “T cells” instead of “t cells”.

 Response to Reviewer

T1DM is now consistently used and ‘t cells’ is changed to ‘T cells’

Be aware of nomenclature for scientific names: Lactobacillus johnsonii, and Candida albicans (italics, genus starts with capital letter) throughout the document and in Figure 2.

 Response to Reviewer

Thank you. Italics and capitalization now consistent and appropriate.

There are several minor mistakes and typos. Here I indicate some of them but there may be more. Please revise the entire document. It would be helpful including line numbers. 

Response to Reviewer

Apologies. The manuscript has now been fully spell checked.

Section 2, 3rd line, “…pancreatic B-cells remain open…”

Response to Reviewer

This has now been pluralized. Thank you.

Section 3.1 7th line, I believe it should say “…by endogenous and exogenous ligands…”; line 12: “…cells and tissues…”; 4th paragraph “YY1 ko” what does ko stand for? Or it may be a typo?; 5th paragraph: …”would indicate…”

Response to Reviewer

The highlighted typos have now been changed. Thank you.

Section 3.2: “…which may be contributed to by alterations in the gut microbiome.” Please correct the sentence.

Response to Reviewer

This has now been changed to read: “…which may also involve alterations in the gut microbiome.”

Figure 1: change “Osidative stress” for “Oxidative stress”

Response to Reviewer

Thank you for spotting this, which has now been changed.

Reviewer 2 Report

The paper is not properly organized and written. Illustrations can add more information to the paper which are not adequate for this study. 

Novelty of this review is not established. The paper is not suitable for publication.

Author Response

Journal: IJMS (ISSN 1422-0067)

Manuscript ID: ijms-2168614       Type: Review

Title:  Type I diabetes pathoetiology and pathophysiology: Role of suppressed Akkermansia muciniphila, Lactobacillus johnsonii, shikimate pathway, and butyrate in the gut microbiome in driving dysregulated intercellular interactions in the pancreas and the ‘bystander’ activation of memory CD8+ T cells.

Author: George Anderson      Section: Molecular Endocrinology and Metabolism

Special Issue: Central and Peripheral Molecular Mechanisms of Metabolism Regulation 2.0

Response to Reviewers

Reviewer 2

The paper is not properly organized and written. Illustrations can add more information to the paper which are not adequate for this study. 

Response to Reviewer

A number of changes, including the addition of 3 new figures, should make the submission more reader-friendly.

Novelty of this review is not established. The paper is not suitable for publication.

Response to Reviewer

The manuscript is reviewing T1DM pathophysiology and highlighting how alterations in the mitochondrial melatonergic pathway may be crucial to understanding why pancreatic B-cells are subject to apoptosis. This is not a derived bit of data as it has not been investigated. However, as indicated throughout the manuscript, the incorporation of the mitochondrial melatonergic pathway allows the collation of wider bodies of previously disparate data, indicating its likely importance. This is a review that is striving to inform future research, including prevention treatment targets.

Reviewer 3 Report

Journal: IJMS (ISSN 1422-0067)

Manuscript ID: ijms-2168614

Type: Review

Title: Type I diabetes pathoetiology: pancreatic B-cell effects of YY1, NF-kB and TLR4 desynchronized from mitochondrial melatonergic pathway induction

Author: George Anderson

Section: Molecular Endocrinology and Metabolism

Special Issue: Central and Peripheral Molecular Mechanisms of Metabolism Regulation 2.0

In this review, the author investigates data on the biological underpinnings of T1DM, highlighting the role of the mitochondrial melatonergic pathway in pancreatic β-cells in integrating previously disparate bodies of data on T1DM. 

The topic of the article is of current scientific interest. The subtopics are presented in an ordered manner and is easy to read. The figures are helpful to understand the main points of the analysis. The conclusions followed from the data analyzed.

Author Response

Reviewer 3

In this review, the author investigates data on the biological underpinnings of T1DM, highlighting the role of the mitochondrial melatonergic pathway in pancreatic β-cells in integrating previously disparate bodies of data on T1DM. 

The topic of the article is of current scientific interest. The subtopics are presented in an ordered manner and is easy to read. The figures are helpful to understand the main points of the analysis. The conclusions followed from the data analyzed.

Response to Reviewer

Thank you for these encouraging comments. Hopefully, the manuscript will drive research that will allow the prevention of pancreatic B-cell loss in the early stages of T1DM.

Reviewer 4 Report

Anderson wrote a review manuscript to propose that the mitochondrial melatonergic pathway in T1DM could impact research and treatment. The main idea of the review is interesting. However, it has serious flaws. The following points are suggested to be addressed to improve the manuscript and be considered for publication.

·      Throughout the manuscript, there are several paragraphs without any references. For example, the following sentence “At low/basal levels of blood glucose, the ATP-sensitive potassium (KATP) channels in pancreatic β-cell remain open, leading to maintained membrane hyperpolarization coupled to Ca2+ channel closure, thereby inhibiting insulin secretion by pancreatic β-cells” needs a proper reference. Please, review the manuscript and add the corresponding references.

·      Please briefly describe which is amylin fibrillation.

·      Please write a better connection for the sentence “Amylin, Ca2+ dysregulation, reactive oxygen species (ROS), endoplasmic reticulum alterations and metabolic dysregulation have all been proposed to drive post-translational protein modifications that underpin the autoimmune-like response in T1DM [11]” with the previous one. In this regard, do all these processes the result of amylin fibrillation? Or are they separate ones?

·      Overall, the section 2 2. Classical type 1 diabetes mellitus pathoetiology is disordered. The author tried to make an overview of the pathoetiology of T1DM. However, the paragraphs are disordered and could be better connected to improve reading fluency.

·      In the sentence “The effects of LPS at TLR4 are mediated via the induction of the transcription factors, NF-kB and YY1, as in many other cell types, including immune and glial cells [36]”. List some of the cell types with their corresponding references.

·      Please check through the manuscript the scientific names and unity them in the Binomial Nomenclature rules. For example, Lactobacillus johnsonii is found without a capital letter; some sentences are found in italics, and others are not.

·      In section 3.2, Gut microbiome and T1DM, most of the evidence is focused on Lactobacillus johnsonii. However, there is only one sentence about pathological microbiome “Importantly, the gut microbiome is comprised of not only bacteria but also fungi and viruses, including enteroviruses and bacteriophages, with all these groupings showing changes at the initiation of paediatric T1DM [77,78]”. Please describe the evidence of those changes in the initiation of T1DM. The author is referred to the following original and review papers to enrich his discussion.

Shilo S, Godneva A, Rachmiel M, Korem T, Bussi Y, Kolobkov D, Karady T, Bar N, Wolf BC, Glantz-Gashai Y, Cohen M, Zuckerman-Levin N, Shehadeh N, Gruber N, Levran N, Koren S, Weinberger A, Pinhas-Hamiel O, Segal E. The Gut Microbiome of Adults With Type 1 Diabetes and Its Association With the Host Glycemic Control. Diabetes Care. 2022 Mar 1;45(3):555-563. DOI: 10.2337/dc21-1656. PMID: 35045174.

van Heck JIP, Gacesa R, Stienstra R, Fu J, Zhernakova A, Harmsen HJM, Weersma RK, Joosten LAB, Tack CJ. The Gut Microbiome Composition Is Altered in Long-standing Type 1 Diabetes and Associates With Glycemic Control and Disease-Related Complications. Diabetes Care. 2022 Sep 1;45(9):2084-2094. DOI: 10.2337/dc21-2225. PMID: 35766965.

Zheng P, Li Z, Zhou Z. Gut microbiome in type 1 diabetes: A comprehensive review. Diabetes Metab Res Rev. 2018;34(7):e3043. doi:10.1002/dmrr.3043

·      The word “paediatric” is misspelled; correct it.

·      The author stated in the abstract and the introduction that the gut permeability is pivotal in developing T1DM pathology. However, the mechanism remained unclear. Please discuss how bacterial, fungal, and viral infections dysregulate it.

·      What does the author mean by mitochondrial subtypes? Specify

·      Given the importance for the manuscript on the melatonergic pathway. Briefly describe the mechanism of how melatonin performs its effects. For example, how does it suppress NFKB or induce autophagy/mitophagy?

·      It is suggested to enlarge the fonts of the images. Carefully review the typos and the adequate nomenclature. For example, in Figure 2, there is an additional space between diet and Hsp70 is a protein; therefore, the first letter must be in capital letter.

·      It is suggested to discuss future research as paragraphs and not as bullets. Given that the bullets only indicate questions, there is no discussion nor integration with the previous evidence.

·      Skin symptoms were not discussed in the manuscript, only in future research. Please, add the corresponding evidence and discuss it throughout the manuscript.

·      Levels of prenatal serotonin were not discussed (just as in the above observation).

Author Response

Journal: IJMS (ISSN 1422-0067)

Manuscript ID: ijms-2168614       Type: Review

Title:  Type I diabetes pathoetiology and pathophysiology: Role of suppressed Akkermansia muciniphila, Lactobacillus johnsonii, shikimate pathway, and butyrate in the gut microbiome in driving dysregulated intercellular interactions in the pancreas and the ‘bystander’ activation of memory CD8+ T cells.

Author: George Anderson      Section: Molecular Endocrinology and Metabolism

Special Issue: Central and Peripheral Molecular Mechanisms of Metabolism Regulation 2.0

Response to Reviewers

Reviewer 4

Anderson wrote a review manuscript to propose that the mitochondrial melatonergic pathway in T1DM could impact research and treatment. The main idea of the review is interesting. However, it has serious flaws. The following points are suggested to be addressed to improve the manuscript and be considered for publication.

Response to Reviewer

Thank you for these encouraging comments as well as for your time and consideration of this manuscript, which has improved the manuscript.

  • Throughout the manuscript, there are several paragraphs without any references. For example, the following sentence “At low/basal levels of blood glucose, the ATP-sensitive potassium (KATP) channels in pancreatic β-cell remain open, leading to maintained membrane hyperpolarization coupled to Ca2+ channel closure, thereby inhibiting insulin secretion by pancreatic β-cells” needs a proper reference. Please, review the manuscript and add the corresponding references.

Response to Reviewer

Thank you for highlighting this. This sentence now has a reference. Occasionally, a sequence of sentences are covered by one reference at the end. I have now referenced other collections of such sentences with a reference after the first of these sentences.

  • Please briefly describe which is amylin fibrillation.

Response to Reviewer

The following sentence has now been added: “Amylin fibrillation follows amylin over-production in association with insulin resistance and hyperinsulinemia, which triggers a nucleation-dependent self-assembly of amylin into intracellular or extracellular amyloid deposits [13].”

  • Please write a better connection for the sentence “Amylin, Ca2+ dysregulation, reactive oxygen species (ROS), endoplasmic reticulum alterations and metabolic dysregulation have all been proposed to drive post-translational protein modifications that underpin the autoimmune-like response in T1DM [11]” with the previous one. In this regard, do all these processes the result of amylin fibrillation? Or are they separate ones?

Response to Reviewer

The following has now been added to the prior sentence to better link to the subsequent sentence: ” ……including in relation to the ‘autoimmune’ aspects of T1DM.”

The processes listed have all been proposed to underpin or contribute to the ‘autoimmune’ aspects of T1DM, without any overall conceptualization as to how they may interact in order to do so. It is proposed here that the ‘autoimmune’ aspects of T1DM are intimately linked to alterations in the mitochondrial melatonergic pathway as a core aspect of mitochondrial function.

  • Overall, the section 2 2. Classical type 1 diabetes mellitus pathoetiology is disordered. The author tried to make an overview of the pathoetiology of T1DM. However, the paragraphs are disordered and could be better connected to improve reading fluency.

Response to Reviewer

The sentences in this paragraph are now linked better, whilst a new figure and legend detailing the tryptophan-melatonin pathway have now been added.   

  • In the sentence “The effects of LPS at TLR4 are mediated via the induction of the transcription factors, NF-kB and YY1, as in many other cell types, including immune and glial cells [36]”. List some of the cell types with their corresponding references.

Response to Reviewer

Cell types and references have now been added.

  • Please check through the manuscript the scientific names and unity them in the Binomial Nomenclature rules. For example, Lactobacillus johnsoniiis found without a capital letter; some sentences are found in italics, and others are not.

Response to Reviewer

Thank you for highlighting this. Lactobacillus johnsonii and Candida albicans have now been put in italics and capitalized.

  • In section 3.2, Gut microbiome and T1DM, most of the evidence is focused on Lactobacillus johnsonii. However, there is only one sentence about pathological microbiome “Importantly, the gut microbiome is comprised of not only bacteria but also fungi and viruses, including enteroviruses and bacteriophages, with all these groupings showing changes at the initiation of paediatric T1DM [77,78]”. Please describe the evidence of those changes in the initiation of T1DM. The author is referred to the following original and review papers to enrich his discussion.

Shilo S, Godneva A, Rachmiel M, Korem T, Bussi Y, Kolobkov D, Karady T, Bar N, Wolf BC, Glantz-Gashai Y, Cohen M, Zuckerman-Levin N, Shehadeh N, Gruber N, Levran N, Koren S, Weinberger A, Pinhas-Hamiel O, Segal E. The Gut Microbiome of Adults With Type 1 Diabetes and Its Association With the Host Glycemic Control. Diabetes Care. 2022 Mar 1;45(3):555-563. DOI: 10.2337/dc21-1656. PMID: 35045174.

van Heck JIP, Gacesa R, Stienstra R, Fu J, Zhernakova A, Harmsen HJM, Weersma RK, Joosten LAB, Tack CJ. The Gut Microbiome Composition Is Altered in Long-standing Type 1 Diabetes and Associates With Glycemic Control and Disease-Related Complications. Diabetes Care. 2022 Sep 1;45(9):2084-2094. DOI: 10.2337/dc21-2225. PMID: 35766965.

Zheng P, Li Z, Zhou Z. Gut microbiome in type 1 diabetes: A comprehensive review. Diabetes Metab Res Rev. 2018;34(7):e3043. doi:10.1002/dmrr.3043

Response to Reviewer

The following has been added to the text: “Importantly, the gut microbiome is comprised of not only bacteria, but also fungi and viruses, including enteroviruses and bacteriophages, with all these groupings showing changes at the initiation of pediatric T1DM [77,78]. Investigations on the gut microbiome in T1DM have focussed on changes in gut bacteria, versus control participants, showing elevations Prevotella copri and Eubacterium siraeum, with relative attenuation of Firmicutes bacterium and Faecalibacterium prausnitzii [79]. Other studies have investigated a wider range of changes in the gut microbiome, indicating no significant differences in α-diversity between T1DM and controls [80]. However, these authors found T1DM patients to have 43 bacterial taxa significantly depleted and 37 bacterial taxa significantly enriched [80]. This study also found disease duration and glycated hemoglobin (HbA1c) to explain a significant part of the gut microbiome variation in T1DM, whilst neuropathy and macrovascular complications were significantly linked to variations in several microbial species [80]. However, as noted by the authors of these studies [79,80], and other studies [81], the mechanistic links to pancreatic β-cell loss and wider T1DM pathophysiology remain to be determined.

Limited data on bacteriophages in the pathoetiology of T1DM indicate that amyloid-producing Escherichia coli (E. coli), E. coli phages, and bacteria-derived amyloid may be involved in the early stages of T1DM pathoetiology, as indicated in data derived from children at high risk of T1DM [82]. This study and other data indicate that changes in the gut virome may precede the initial signs of T1D, and therefore be of relevance to T1DM pathoetiology [83]. The causative relationship has still to be determined. However, such data may indicate that the gut virome may be more relevant to T1DM pathoetiology than gut bacteria, which tend to show diversity only after the emergence of T1DM. Enteroviruses are one of the major environmental triggers of childhood-onset T1DM, with recent data indicating that enteroviruses may also be an important trigger in adult-onset T1DM [84]. Overall, data indicate an interrelatedness of the gut microbiota, metaproteome and virome that is relevant to T1DM onset, as investigated in young children, with a  functional remodeling of the gut microbiota accompanying islet autoimmunity [77]. An initial bacteriophage or enterovirus impact on gut microbiome diversity seems followed by a decrease in butyrate-producing bacteria, with consequences for mitochondrial function systemically. As to whether the suppressed butyrate and/or other gut bacteria products are drivers of changes in pancreatic β-cells, either via directly or indirectly suppressing the mitochondrial melatonergic pathway in pancreatic β-cells requires investigation.

The above highlights processes that form the pathophysiology of T1DM and the importance of factors dysregulating the mitochondrial melatonergic pathway in pancreatic β-cells. However, the pathoetiology of T1DM is still the subject of intense investigation. Recent work has highlighted alterations in the wider gut microbiome, including significant roles for enteroviruses and bacteriophages in the pathogenesis and pathophysiology of T1DM [77,78], ultimately resulting in changes in gut bacteria and suppression of the short-chain fatty acids, such as butyrate. As noted, recent work has indicated pathophysiological overlaps of amyotrophic lateral sclerosis (ALS) and T1DM [4], with ALS showing significant changes in the gut microbiome, including as induced by glyphosate-based herbicides (GBH), which suppress the shikimate pathway [7]. In humans and animals the shikimate pathway is primarily achieved by Akkermansia muciniphila [150]. In poultry, Akkermansia muciniphila is significantly suppressed by GBH, and does not seem to be restored following GBH cessation [151]. Preclinical data shows GBH to suppress butyrate and propionate [151] and to bias the thriving of some bacteria, as indicated by an increase in α-diversity [152]. Interestingly, Akkermansia muciniphila is a significant regulator of T1DM pathoetiology [153], with the transfer of Akkermansia muciniphila remodelling the gut microbiome, maintaining the gut barrier, reducing circulating LPS and TLR expression, as well as reducing pancreatic islet autoimmunity and significantly delaying T1DM onset in non-obese diabetic (NOD) mice [154]. In T1DM patients, Akkermansia muciniphila levels  negatively correlate with glucose level and HbA1c [155], indicating a significant association with T1DM pathophysiology. Probiotic administration, including of Lactobacillus johnsonii, increases Akkermansia muciniphila in T1DM patients, in association with improved glucose control and HbA1c levels [156]. Interactions of Akkermansia muciniphila with bacteriophages can regulate Akkermansia muciniphila levels [157], indicating important bacteriophage impacts via Akkermansia muciniphila and the shikimate pathway.

The suppression of the shikimate pathway decreases not only tryptophan but also tryptophan-derived ligands for the AhR, such as tryptamine and indole-3-acetate, thereby indicating an association of a suppressed shikimate pathway not only with gut permeability, but also with the regulation of NK cell and CD8+ T cell cytotoxicity [38]. The suppressed levels of Akkermansia muciniphila in T1DM may therefore be intimately linked to lower tryptophan (as well as tyrosine and phenylalanine), increased gut permeability and an altered capacity of the gut microbiome to suppress NK cells and CD8+ T cells. Although AhR activation by kynurenine can induce a state of ‘exhaustion’ in NK cells and CD8+ T cells [38], the AhR has differential effects in memory CD8+ T cells, with AhR activation suppressing circulating memory CD8+ T cells, while promoting the core gene program of resident memory CD8+ T cells [158]. As the gut is proposed to be an important site for the inappropriate maintenance of autoreactive memory CD8+ T cells, whereby autoimmune-linked autoreactive, memory CD8+ T cells may interact in the Peyer’s patches of the gut to escape thymic deselection [Okada et al., 2023], the effects of enteroviruses and bacteriophages in T1DM pathoetiology may be mediated via suppressed Akkermansia muciniphila levels and shikimate pathway activity. The activation of autoreactive T cells in the gut seems via bystander activation and not from "molecular mimicry" arising from cross-reactivity between gut microbiota-derived peptides and islet-derived epitopes [159]. In a preclinical T1DM model, these authors show that the initial activation of islet-specific CD8+ T cells occurs in the pancreatic lymph nodes but gain additional effector function in the gut lymphoid tissues via non-specific bystander activation [159]. These authors also showed that the oral administration of the short-chain fatty acid, butyrate, attenuated the ‘bystander’ induced cytotoxic effector functions of these autoreactive CD8+ T cells [159].

This requires further investigation as it would indicate that initial changes occurring in the intercellular interactions of pancreatic islet cells drive alterations in the mitochondrial melatonergic pathway in pancreatic β-cells that decreases PINK1/parkin/LETM1/mitophagy and increases MHC-1 that primes CD8+ T cells that would normally be subject to thymic deselection. However, the changes in the wider gut microbiome, including as induced by bacteriophages, enteroviruses and possibly GBH, suppress Akkermansia muciniphila and the shikimate pathway to provide a gut microenvironment, possibly in Peyer’s patches and involving variations in different AhR ligands and levels, which allows these autoreactive CD8+ T cells to escape thymic deselection. This would indicate that alterations in the gut microbiome are relevant to different aspects of T1DM pathoetiology and pathophysiology, namely changes in intercellular interactions of cells in pancreatic islets as well as in the maintenance of memory CD8+ T cells arising from the intercellular interactions driving changes in the mitochondrial melatonergic pathway of pancreatic β-cells.

Along with the 3 recommended references and other references.

Also the following has been added to the Future Research Section:

T1DM is associated with an increased risk of amyotrophic lateral sclerosis (ALS) in people aged < 50years of age [156], with streptozotocin-induced T1DM also leading to neuromuscular junction retraction and muscle atrophy [157]. There also seems genome-wide genetic overlaps between T1DM and ALS [158], whilst glyphosate-based herbicides, a proposed risk factor for ALS [7], induces a T2DM phenotype when combined with a high fructose diet [159]. Chronic glyphosate causes severe degeneration in pancreatic acinar cells and islets of Langerhans [160]. This could suggest epigenetic and genetic overlaps of ALS and T1DM. Is this mediated via gut, immune and/or mitochondrial melatonergic related factors? Are glyphosate-based herbicides an environmental risk factor for T1DM? Glyphosate-based herbicides can inhibit the shikimate pathway, which is a relevant provider of tryptophan to the body, with the inhibition of the shikimate pathway increasing gut permeability and gut dysbiosis, including decreased butyrate producing gut bacteria [7]. Do enteroviruses and/or bacteriophages in T1DM suppress the shikimate pathway? Do alterations in the gut prime autoreactive CD8+ T cells and prevent their elimination/deselection in the thymus?”

  • The word “paediatric” is misspelled; correct it.

Response to Reviewer

‘Paediatric’ is ‘UK English’, ‘pediatric’ is ‘American English’…..

  • The author stated in the abstract and the introduction that the gut permeability is pivotal in developing T1DM pathology. However, the mechanism remained unclear. Please discuss how bacterial, fungal, and viral infections dysregulate it.

Response to Reviewer

There is a lot to be discovered regarding the gut microbiome and it would be too tentative to detail any differential effects of bacterial, fungal and viral infections in the regulation of the mitochondrial melatonergic pathway in pancreatic B-cells. Alterations in the gut microbiome are detailed throughout the manuscript, based on changes in circulating LPS and short-chain fatty acids, which have a good basis in data. This is the emphasis on the gut microbiome.

  • What does the author mean by mitochondrial subtypes? Specify

Response to Reviewer

This is explained in the following sentence and reference: “These authors showed mitochondrial subtypes could be classed according to mtDNA- and nuclear DNA- encoded OXPHOS genes [89].” I have put subtypes in inverted commas to reflect their use by the authors. I suspect that the ‘subtypes’ described may not survive further investigation. However, the data in this study does emphasize how an array of factors and processes may associate with changes in mitochondrial function. I have added the following at the end of this paragraph: “ As to whether these ‘subtypes’ described survive further investigation remains to be determined, especially when alterations in the regulation of the mitochondrial melatonergic pathway are included. However, the data in this study does emphasize how an array of factors and processes may associate with changes in mitochondrial function.”

  • Given the importance for the manuscript on the melatonergic pathway. Briefly describe the mechanism of how melatonin performs its effects. For example, how does it suppress NFKB or induce autophagy/mitophagy?

Response to Reviewer

The following has been added at the first mention of melatonin in Section 2: “The antioxidant, anti-inflammatory, circadian and mitochondrial optimizing effects of melatonin are important aspects of T1DM, whilst the immediate precursor of melatonin, N-acetylserotonin (NAS) may also be a crucial regulator of pancreatic β-cell survival, via its capacity to activate the brain-derived neurotrophic factor (BDNF) receptor TrkB, as shown in figure 1. The role of the melatonergic pathway in T1DM is detailed in section 5.” A new figure (figure 1) and legend have also been added which detail the melatonergic pathway and melatonin effects. 

  • It is suggested to enlarge the fonts of the images. Carefully review the typos and the adequate nomenclature. For example, in Figure 2, there is an additional space between diet and Hsp70 is a protein; therefore, the first letter must be in capital letter.

Response to Reviewer

Thank you. Typos have now been corrected and fonts in the figures have been enlarged.

  • It is suggested to discuss future research as paragraphs and not as bullets. Given that the bullets only indicate questions, there is no discussion nor integration with the previous evidence.

Response to Reviewer

Bullets are preferred, as they are discreet and not tainted/limited by an overarching perspective.

  • Skin symptoms were not discussed in the manuscript, only in future research. Please, add the corresponding evidence and discuss it throughout the manuscript.

Response to Reviewer

Although skin symptoms are clinically relevant, this is too tangential to the main theme of the manuscript and could overload an already data-dense manuscript. However, given the data on the role of melatonin in the skin, coupled to the presence, and functional relevance, of the melatonergic pathway in skin cells, it will be interesting to clarify in future research.

  • Levels of prenatal serotonin were not discussed (just as in the above observation).

Response to Reviewer

As with the association of gestational diabetes it is necessary for future research to look at the role of prenatal serotonin as a precursor for the mitochondrial melatonergic pathway in an array of different cells, including the placenta as well as in the developing foetus. I think this would be too tentative to detail currently, with its relevance determined by future data, although may ultimately be very important. However, the following as been added to the Future Research Section: “Alterations in tryptophan and serotonin are evident in gestational diabetes [156], indicating consequences for the mother, placenta and offspring [157], including in the regulation of the mitochondrial melatonergic pathway. This will be important to clarify in future research.”

Author Response

Journal: IJMS (ISSN 1422-0067)

Manuscript ID: ijms-2168614       Type: Review

Title:  Type I diabetes pathoetiology and pathophysiology: Role of suppressed Akkermansia muciniphila, Lactobacillus johnsonii, shikimate pathway, and butyrate in the gut microbiome in driving dysregulated intercellular interactions in the pancreas and the ‘bystander’ activation of memory CD8+ T cells.

Author: George Anderson      Section: Molecular Endocrinology and Metabolism

Special Issue: Central and Peripheral Molecular Mechanisms of Metabolism Regulation 2.0

Reviewer 5

The manuscript is a comprehensive review covering the pathoetiology and pathophysiology of Type I diabetes, particularly focusing on the role of the melatonergic pathway in preserving pancreatic b-cell mitochondrial metabolism and function.

The author nicely guides the reader through the most recent available literature on the topic, and the manuscript is well-structured. However, I would recommend some minor adjustments to the following parts:

Response to Reviewer

Thank you for these encouraging comments.

  • There is a lack of general introduction on melatonin itself as a key hormone with major role in the sleep-wake cycle. Even though the function of melatonin is of general knowledge and it is a well-known hormone for researchers in the field, I believe all readers should be introduced to it before mentioning the melatonergic pathway.

Response to Reviewer

Figure 1 showing the tryptophan-melatonin pathway has now been added at the end of Section 2, at the first mention of melatonin and the melatonergic pathway. As well as the figure and legend, the following text has also been added at this point:

The antioxidant, anti-inflammatory, circadian and mitochondrial optimizing effects of melatonin are important aspects of T1DM, whilst the immediate precursor of melatonin, N-acetylserotonin (NAS) may also be a crucial regulator of pancreatic β-cell survival, via its capacity to activate the brain-derived neurotrophic factor (BDNF) receptor TrkB, as shown in figure 1. The role of the melatonergic pathway in T1DM is detailed in section 5.”

  • The initial paragraphs (numbers 1 to 4) contain sentences referring to the melatonergic pathway, a complex signalling cascade of events that is described in a later paragraph (number 5). For clarity, I believe melatonin and the melatonergic pathway should be introduced and described earlier in the manuscript, so that all sentences referring to it will be easier to be put into context by the reader.

Response to Reviewer

Figure 1 showing the tryptophan-melatonin pathway has now been added at the end of Section 2, at the first mention of melatonin and the melatonergic pathway. As well as the figure and legend, the following text has also been added at this point:

The antioxidant, anti-inflammatory, circadian and mitochondrial optimizing effects of melatonin are important aspects of T1DM, whilst the immediate precursor of melatonin, N-acetylserotonin (NAS) may also be a crucial regulator of pancreatic β-cell survival, via its capacity to activate the brain-derived neurotrophic factor (BDNF) receptor TrkB, as shown in figure 1. The role of the melatonergic pathway in T1DM is detailed in section 5.”

  • Paragraph 3.2 seems to belong to paragraph 3.1 (gut dysbiosis/permeability is listed among the pathophysiological processes and factors in T1DM). The author may perhaps combine the two paragraphs instead.

Response to Reviewer

Paragraphs 3.1 and 3.2 have now been combined, as suggested.

  • The section in paragraph 3.1 referring to YY1 KO murine models contains reference #31, but the reference is actually #32.

Response to Reviewer

Thank you for spotting this, which has now been corrected.

  • Paragraphs 5 and 5.1 may possibly be re-structured so that 5.1 may describe the melatonergic pathway, and 5.2 may describe T1DM pathophysiology (instead of having melatonergic pathway and T1D twice in titles, causing some redundancies).

Response to Reviewer

I understand the rationale of what you propose. However, I am attempting to emphasize how the incorporation of the melatonergic pathway can link both classical and wider aspects of T1DM pathophysiology, so there is some appeal in emphasizing that by maintaining these sub-headings.

  • Figure 1 should also show acetyl-CoA as a necessary substrate for AANAT.

Response to Reviewer

Thank you. This has now been included in the figure and legend.

  • Similarly to LPS and butyrate, the origin of HMGB1 and hsp70 may be indicated in Figure 2, as in the schematic they look a bit out of context. Are these two factors induced by gut dysbiosis/permeability (as LPS and butyrate)? If not, perhaps a “stress”-labelled box may be introduced as their source?

Response to Reviewer

Thank you. A local Stress/inflammation box has now been included and the legend modified.

  • The last two sections of paragraph 6, as well as the whole paragraph 8, pose a lot of questions for the reader. Even though I understand that several questions on this topic remain unanswered by the available literature, I find this approach to be very confusing for the reader. Perhaps the author may limit the “opened questions” section to paragraph 8 only, while rephrasing sentences in paragraph nr 6 so that the reader is informed on the existing literature on the topic that rather provide knowledge in the field.

Response to Reviewer

I agree. The questions formerly posed at the end of Section 6 have now been omitted and replaced by reviewed literature, where appropriate.

Round 2

Reviewer 1 Report

In this revised version of the article the author answered all my questions, the figures are helpful, and the effort is appreciated.

I would just suggest a shorter (more general) title (for example “Type I diabetes pathoetiology and pathophysiology: implications of gut microbiome, pancreatic cellular interactions, and the ‘bystander’ activation of memory CD8+ T cells”). It is up to the author.

Author Response

Manuscript ID: ijms-2168614       Type of manuscript: Review

Title: Type I diabetes pathoetiology and pathophysiology: roles of the gut microbiome, pancreatic cellular interactions, and the ‘bystander’ activation of memory CD8+ T cells

Author: George Anderson

Response 2 to Reviewer 1:

In this revised version of the article the author answered all my questions, the figures are helpful, and the effort is appreciated.

I would just suggest a shorter (more general) title (for example “Type I diabetes pathoetiology and pathophysiology: implications of gut microbiome, pancreatic cellular interactions, and the ‘bystander’ activation of memory CD8+ T cells”). It is up to the author.

Response to reviewer:

Thank you for your encouraging comments and suggestion. The title of the manuscript is now changed to: “Type I diabetes pathoetiology and pathophysiology: roles of the gut microbiome, pancreatic cellular interactions, and the ‘bystander’ activation of memory CD8+ T cells”

Reviewer 2 Report

Changes are acceptable and giving sound knowledge about research idea.

Author Response

Manuscript ID: ijms-2168614       Type of manuscript: Review

Title: Type I diabetes pathoetiology and pathophysiology: roles of the gut microbiome, pancreatic cellular interactions, and the ‘bystander’ activation of memory CD8+ T cells

Author: George Anderson

Response 2 to Reviewer 2

Changes are acceptable and giving sound knowledge about research idea.

Response to reviewer:

Thank you for these encouraging comments.

Reviewer 4 Report

In this version, Anderson extended his discussion on the review and replied with detail of my previous comments. Some minor comments are highlighted.

In the previous revision, the author was encouraged to enlarge and higher-resolution images. Figure 1, and Figure 3 still need a better resolution.

Check the whole Figure 4. Some shapes overlap with words. Glutamate is incorrectly written (glutamae)

Author Response

Manuscript ID: ijms-2168614       Type of manuscript: Review

Title: Type I diabetes pathoetiology and pathophysiology: roles of the gut microbiome, pancreatic cellular interactions, and the ‘bystander’ activation of memory CD8+ T cells

Author: George Anderson

Response 2 to Reviewer 4

In this version, Anderson extended his discussion on the review and replied with detail of my previous comments. Some minor comments are highlighted.

 Response to Reviewer 4:

Thank you for these encouraging comments.  

In the previous revision, the author was encouraged to enlarge and higher-resolution images. Figure 1, and Figure 3 still need a better resolution.

 Response to Reviewer 4:

The font in Figure 1 and Figure 3 have been increased.

Check the whole Figure 4. Some shapes overlap with words. Glutamate is incorrectly written (glutamae)

 Response to Reviewer 4:

Thank you for spotting this, which has now been corrected